# SHR and SCR coordinate root patterning and growth early in the cell cycle

Cara M. Winter[1,2,4 ✉], Pablo Szekely[1,2,4 ✉], Vladimir Popov[1], Heather Belcher[1], Raina Carter[1], Matthew Jones[3], Scott E. Fraser[3], Thai V. Truong[3] & Philip N. Benfey[1,2]

Precise control of cell division is essential for proper patterning and growth during the development of multicellular organisms. Coordination of formative divisions that generate new tissue patterns with proliferative divisions that promote growth is poorly understood. SHORTROOT (SHR) and SCARECROW (SCR) are transcription factors that are required for formative divisions in the stem cell niche of *Arabidopsis* roots[1,2]. Here we show that levels of SHR and SCR early in the cell cycle determine the orientation of the division plane, resulting in either formative or proliferative cell division. We used 4D quantitative, long-term and frequent (every 15 min for up to 48 h) light sheet and confocal microscopy to probe the dynamics of SHR and SCR in tandem within single cells of living roots. Directly controlling their dynamics with an *SHR* induction system enabled us to challenge an existing bistable model[3] of the SHR–SCR gene-regulatory network and to identify key features that are essential for rescue of formative divisions in *shr* mutants. SHR and SCR kinetics do not align with the expected behaviour of a bistable system, and only low transient levels, present early in the cell cycle, are required for formative divisions. These results reveal an uncharacterized mechanism by which developmental regulators directly coordinate patterning and growth.

The final size, shape and function of tissues in multicellular organisms hinge upon the precise control of cell division[4]. Owing to intrinsic and extrinsic cell polarity, a 90° rotation of the division plane determines whether a cell will divide formatively (producing daughter cells with different fates) or proliferatively[5,6] (producing daughter cells with similar fates). A wrong choice can lead to over-proliferation of cells, resulting in aberrant morphogenesis or tumorigenesis[7,8]. Developmental regulators that specify cell fate and interface directly with the cell cycle machinery[9–11] are likely arbiters of this decision. However, we have limited knowledge about how these regulators dynamically control cell division in situ.

SHR and SCR control the formative division in the *Arabidopsis* root that gives rise to the endodermis and cortex cell types (ground tissue). SHR, a mobile intercellular signalling molecule, is produced in the central tissues of the root and moves outward into adjacent cells, including the endodermis and the cortex–endodermal initial daughter (CEID) cell, where it activates *SCR* expression[12,13]. SHR and SCR together activate the expression of the cell cycle regulator *CYCLIND6;1* (*CYCD6*) only in the CEID, triggering formative division[14]. In *shr* and *scr* mutants, this division does not occur, resulting in a single ground tissue mutant layer, rather than distinct files of endodermis and cortex cells[1,2,15] (Fig. 1a).

Cruz-Ramírez et al.[3] proposed a bistable model to explain both how and where SHR and SCR trigger the decision to divide. According to the model, two positive feedback loops generate high stable steady states of SCR and nuclear SHR, triggering formative division (Fig. 1b). Bistability arising from positive feedback is at the heart of mathematical models of decision making in many systems[16]. However, positive feedback does not always lead to bistability[17], and alternative decision-making mechanisms exist. For example, the simple presence of a factor at the right place and time can alter the cell cycle programme and lead to a different cell fate[11].

Quantitative time-lapse imaging of transcription factor dynamics has provided key insights into gene-regulatory network function in single cell organisms and mammalian cell lines[18–20]. Assays of multiple transcription factors in tandem on a long timescale can enable examination of their regulatory relationships[21]. However, many technical challenges have made studies of network dynamics in vivo difficult[22]. Phototoxicity and photobleaching, in particular, restrict studies using confocal microscopy to short timescales or infrequent sampling and limit the number of fluorophores that can be imaged simultaneously. Owing to its lower phototoxicity, light sheet microscopy provides the means for longer-term multi-colour imaging of protein dynamics in vivo. This potential has been extolled for nearly two decades, but the technology has been used primarily for observation of cellular dynamics and morphology changes during development[23–25].

Here, we use long-term 4D imaging of living roots and quantitative analysis to gain insight into the dynamics of the SHR–SCR gene-regulatory network that controls formative divisions in the root stem cell niche. Our measurements revealed a key aspect that

[1]Department of Biology, Duke University, Durham, NC, USA. [2]Howard Hughes Medical Institute, Duke University, Durham, NC, USA. [3]Translational Imaging Center, Bridge Institute, University of Southern California, Los Angeles, CA, USA. [4]These authors contributed equally: Cara M. Winter, Pablo Szekely. ✉e-mail: cara.winter@duke.edu; caraw97@gmail.com; pablo.szekely@duke.edu; plinkush@gmail.com

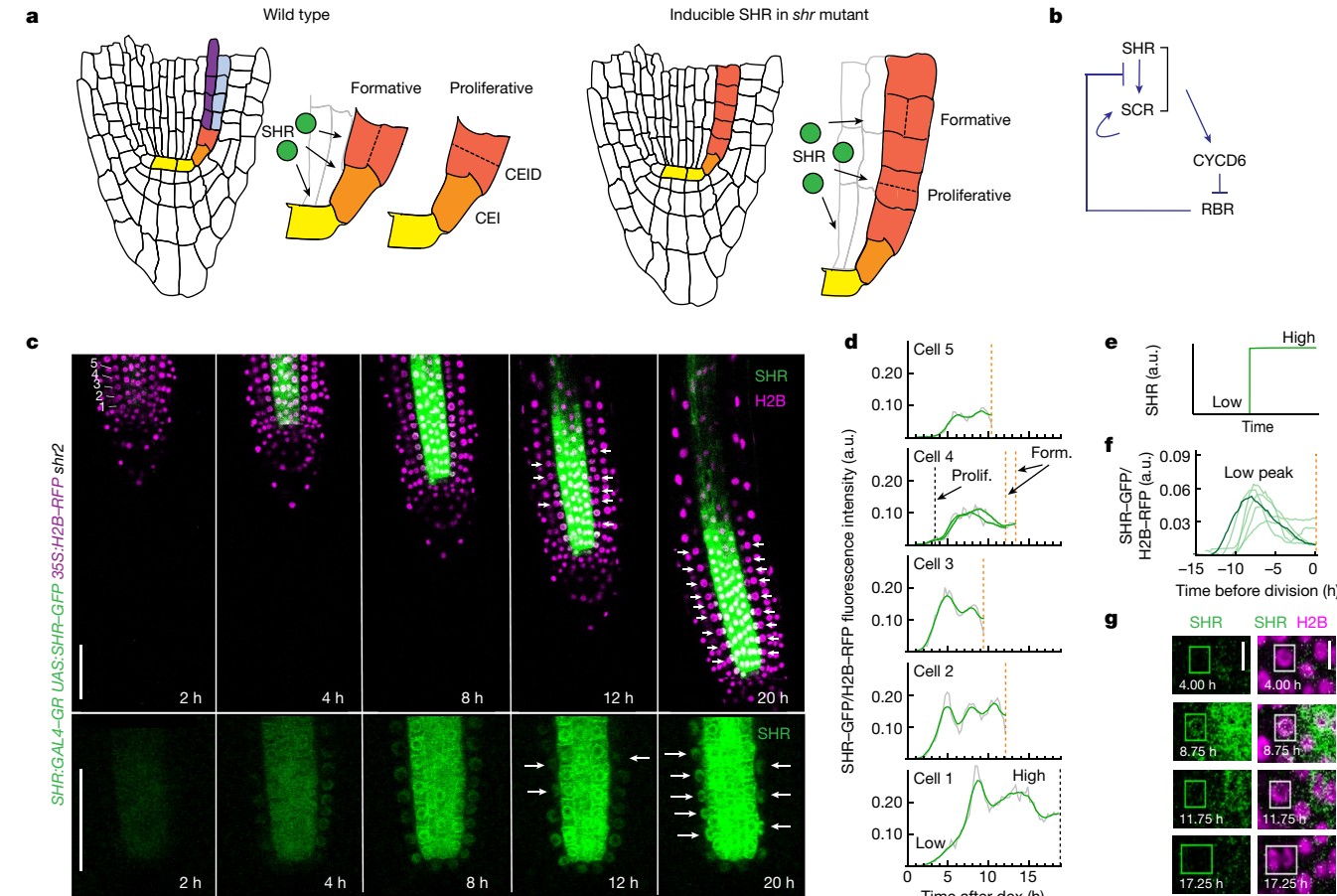

**Fig. 1 | Long-term 4D confocal imaging of SHR reveals dynamics inconsistent with bistability. a**, Diagram of *Arabidopsis* wild-type and *SHR:GAL4–GR UAS:SHR–GFP shr2* mutant roots showing proliferative and formative division planes (adapted from ref. 52). SHR moves from the central tissues of the root into the adjacent cell layer. *SHR* expression and formative divisions occur in the inducible line upon treatment with dex. Yellow, QC (quiescent centre); orange, CEI (cortex–endodermal initial); red, CEID (cortex–endodermal initial daughter) and *shr* mutant layer; blue, cortex; purple, endodermis. **b**, Diagram of the SHR–SCR regulatory network controlling formative division based on Cruz–Ramirez et al.[3]. **c**, Confocal median longitudinal sections showing GFP-labelled SHR and H2B–RFP at timepoints after induction with 10 μM dex. Images are representative of independent timecourse experiments with eight roots. Numbers at the top left show the first five cell positions in the mutant ground tissue. Gamma is set to 0.75 to show signal in the mutant layer for the GFP-only images. Top and bottom show different roots. White arrows,

formative divisions. Scale bars, 50 μm. **d**, Raw (grey) and smoothed (green) SHR trajectory (SHR-GFP/H2B–RFP fluorescence intensity) over time in the first five cells of a single cell file after full induction (10 μM dex). Plots are representative of 211 cells from independent time courses with 8 roots. Possible low and high steady states are indicated for cell 1. Black dashed line, proliferative division; orange dashed line, formative division. a.u., arbitrary units. **e**, SHR trajectory predicted by the Cruz–Ramirez model showing low and high steady states. **f**, SHR trajectories for cells that show a low peak of SHR accumulation hours prior to dividing formatively. Roots were treated with low dex (0.02 μM or 0.03 μM). Dark green, SHR trajectory corresponding to images in **g**. **g**, Median longitudinal sections through a root tip treated with low dex (0.02 μM) highlighting a cell with a low transient peak of SHR prior to dividing formatively. Plots and images in **f** and **g** are representative of 15 cells from 10 roots showing similar behaviour. Scale bars, 10 μm.

was missing from the existing model: namely, that SHR and SCR levels are interpreted within the context of the cell cycle. We present evidence that low threshold levels of SHR and SCR are sufficient and act early in the cell cycle to change the orientation of the division plane.

## SHR dynamics determine formative division

To investigate the mechanism by which SHR dynamically controls formative cell divisions, we generated a fluorescently labelled inducible SHR construct, *SHR:GAL4–GR UAS:SHR–GFP*, capable of rescuing the formative divisions absent in *shr2* mutants[1,2] (Fig. 1a). We induced transcription of *SHR–GFP* in its endogenous expression domain (the central tissues of the root; Fig. 1a) using diminishing concentrations of dexamethasone[26] (dex) (10, 1, 0.05, 0.03, 0.02 and 0.01 μM), and imaged the roots in 3D every 15 min for up to 24 h using confocal microscopy (Fig. 1c and Supplementary Videos 1–4). We observed movement of SHR-GFP into the adjacent mutant ground tissue nuclei, as predicted

from previous SHR localization studies[27,28]. We then quantified the fluorescence of SHR–GFP in the ground tissue nuclei (*n* = 935 cells from 29 roots) relative to a nuclei marker as a measure of SHR concentration (Extended Data Fig. 1a and Supplementary Methods), from the time of induction up to formative division or the end of the experiment if no division occurred (Fig. 1d and Supplementary Data 1).

The inducible SHR line enabled us to produce data from many cells in each root, and to produce a variety of SHR accumulation trajectories with different division outcomes (Fig. 1d and Extended Data Fig. 1b–f). We observed near complete rescue of meristematic formative divisions at 1 μM and 10 μM dex (96% and 99%, respectively) and no formative divisions at a concentration of 0.01 μM dex (Extended Data Fig. 1f). Occasionally, a cell divided anticlinally (proliferatively) before the periclinal formative division (Fig. 1d, cell 4). The rescued formative divisions are likely to be controlled by the SHR–SCR–CYCD6 pathway that controls CEID division in the stem cell niche of wild-type plants (Extended Data Fig. 2a–d and Supplementary Note 1), and levels of

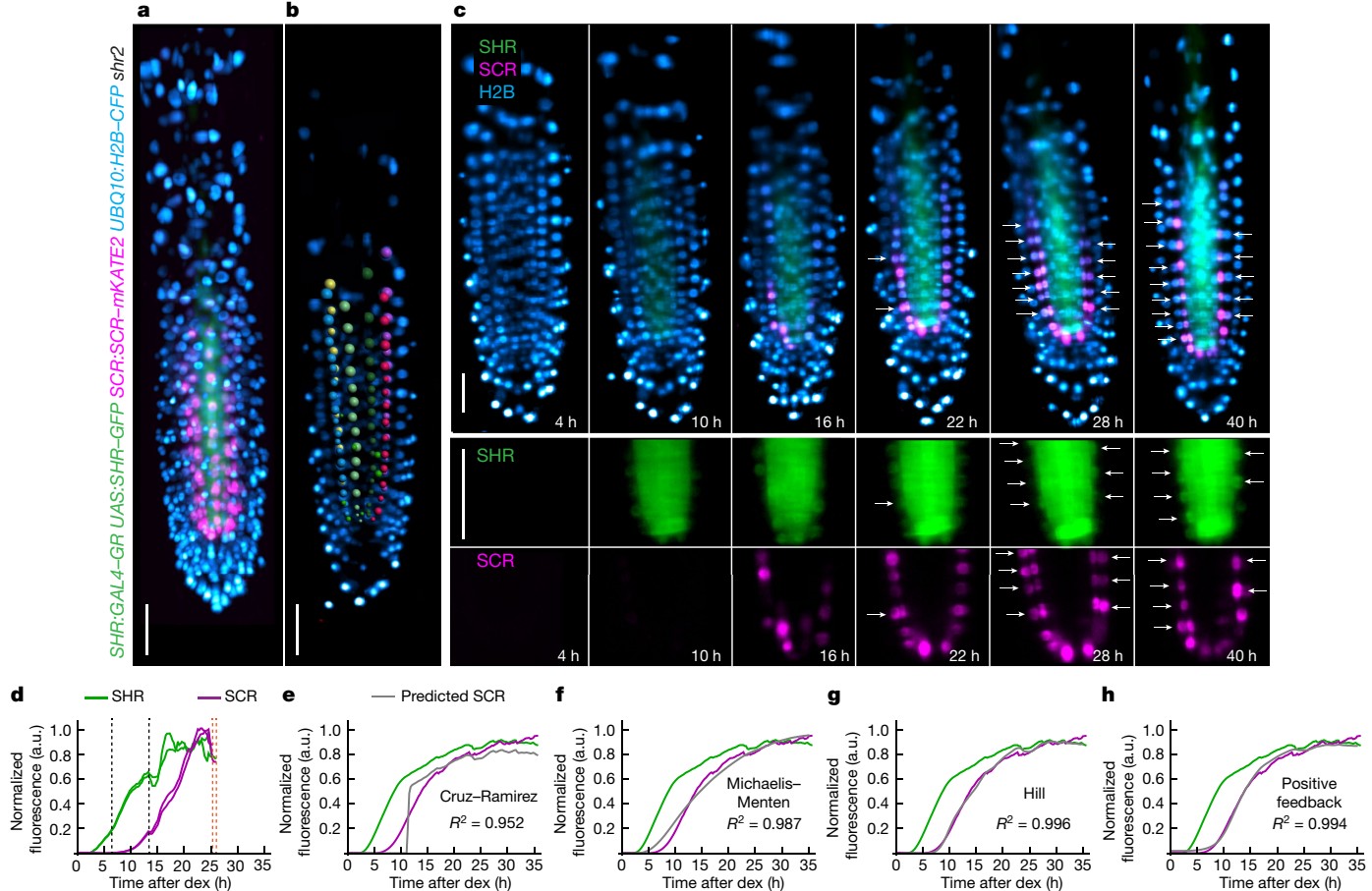

**Fig. 2 | Long-term 4D light sheet imaging of SHR and SCR dynamics reveals that bistability is not required to model their regulatory relationship. a**, 3D reconstruction of a *z*-stack showing induced SHR–GFP (green), SCR–mKATE2 (magenta) and H2B–CFP (blue) fluorescence in a *SHR:GAL4-GR UAS:SHR–GFP SCR:SCR–mKATE2 UBQ10:H2B–CFP shr2* root. Scale bar, 50 µm. **b**, Endodermal nuclei detected in Imaris are selected for quantification. Colours specify different cell files. Scale bar, 50 µm. **c**, Median longitudinal sections of the root in **a** and **b** showing timepoints after induction with 40 µM dex. Images in **a**–**c** are representative of independent timecourse experiments with nine roots. White arrows, formative divisions. Scale bars, 50 µm. **d**, Quantification of SHR and SCR trajectories (SHR–GFP/H2B–CFP and SCR–mKATE2/H2B–CFP fluorescence

intensity, respectively) for a single cell after full induction (40 µM dex). Measurement is representative of 274 cells from independent timecourse experiments with nine roots. SHR and SCR trajectory values are normalized to the 90th quantile (Supplementary Methods). Black dashed line, proliferative division; orange dashed line, formative division. **e**–**h**, Mean of all fully induced SHR (green) and SCR (magenta) normalized trajectories (Supplementary Methods) and predictions for SCR (grey lines) from the Cruz-Ramirez (**e**), Michaelis–Menten (**f**), Hill (**g**) and positive feedback (**h**) models. $R^2$, adjusted *R* squared; *n* = 274 cells from 9 roots (treated with 40 µM dex; Supplementary Methods).

SHR–GFP in fully induced (10 µM dex) plants were comparable to levels of *SHR:SHR–GFP* in the relevant tissues (Extended Data Fig. 2e,f).

The bistable model[3] postulates that SHR triggers formative division when nuclear SHR levels in the ground tissue flip to a high steady state (Fig. 1e). Consistent with this model, in many cases we observed a rapid increase in SHR levels followed by a period during which higher SHR levels were relatively constant prior to division (Fig. 1d, cell 2). However, in other cases, a transient low peak of SHR was able to trigger division many hours later (Fig. 1f,g, Extended Data Fig. 1b and Supplementary Video 5). These SHR kinetics are inconsistent with a bistable model in which high steady state levels of nuclear SHR are necessary to trigger division, where the scale of the model (predicted SHR levels) is comparable to the observed range of SHR protein levels (Fig. 1e–g). We sought next to directly examine SHR regulation of SCR levels.

## Bistability is not required for SHR regulation of SCR

We measured SHR and SCR accumulation simultaneously in single nuclei, using a custom light sheet microscope built to image growing root tips under near physiological conditions[29] (Extended Data

Fig. 3a–c, Supplementary Video 6 and Supplementary Methods). We first introduced *SCR:SCR–mKATE2* into the *SHR:GAL4–GR UAS:SHR–GFP shr2* background. Next, we induced transcription of *SHR–GFP* and subsequent activation of *SCR–mKATE2* expression using different concentrations of dex (40 µM, 20 µM and 0.4 µM; Supplementary Methods) to obtain a range of SHR and SCR accumulation profiles. We imaged and quantified the levels of SHR and SCR in the ground tissue nuclei every 15 min for up to 48 h after induction (Fig. 2a–d, Supplementary Videos 7 and 8 and Supplementary Data 2; *n* = 577 cells from 14 roots). In fully induced roots (40 µM dex), 89% of the meristematic cells had divided after 45 h.

SHR and SCR levels appeared to follow simple dynamics (Fig. 2d). The average SHR and SCR curves closely follow a sigmoid pattern, with SCR having a slightly steeper and later rise (Fig. 2e–h). To further investigate the regulation of SCR by SHR, we fit the data to the bistable model[3] and to three basic ODE models (Supplementary Methods). For each model, we used the averaged SHR dynamics as input to predict SCR expression. Using the published parameters[3], the bistable model predicted a rapid jump in SCR levels that we did not observe in the data (Fig. 2e). By varying each parameter by two orders of magnitude, we

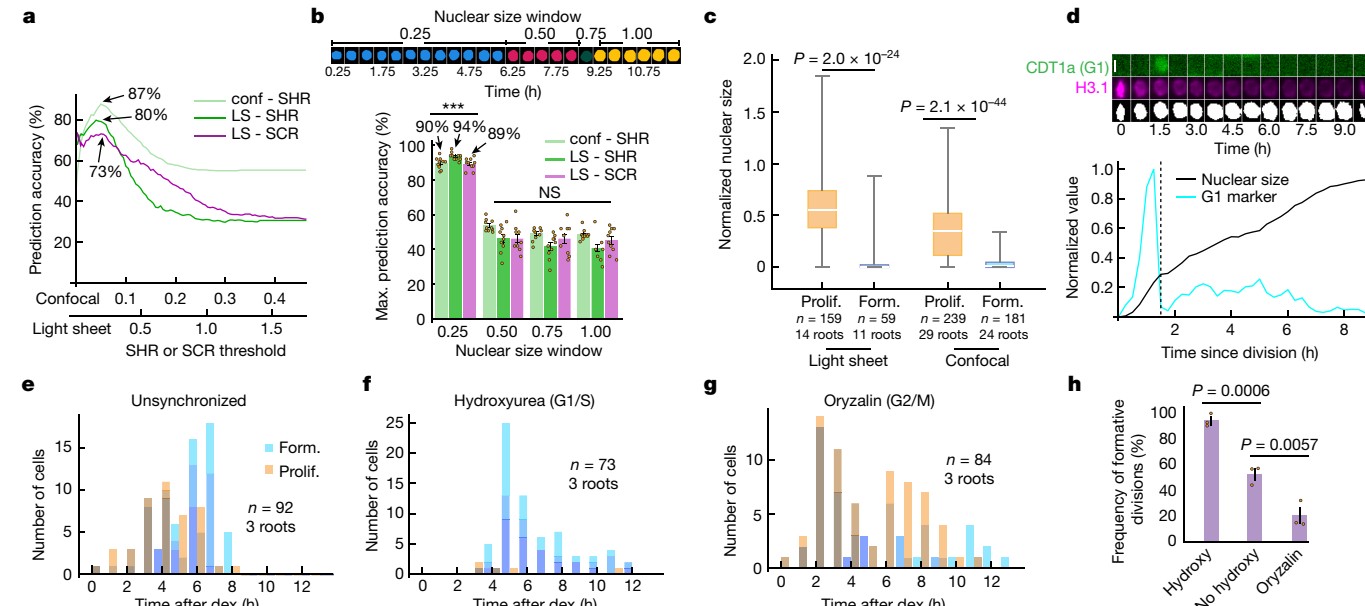

**Fig. 3 | Low threshold levels of SHR and SCR present during an early cell cycle window specify formative division. a**, Prediction accuracy of trajectory classification into formatively dividing and non-dividing cells for a range of SHR and SCR thresholds (Supplementary Methods). Light sheet (LS), $n = 449$ cells from 14 roots; confocal (conf), $n = 743$ cells from 29 roots. **b**, Maximum prediction accuracy of trajectory classification into proliferatively and formatively dividing cells for a given nuclear size window. ***$P = 3.8 \times 10^{-91}$, $8.9 \times 10^{-61}$, $8.0 \times 10^{-43}$ for conf - SHR, LS - SHR and LS - SCR, respectively; one-tailed binomial test. An approximate 50% accuracy is expected by chance. Light sheet, $n = 500$ cells from 14 roots; confocal, $n = 633$ cells from 29 roots. Top, example masks used to calculate the nuclear size trajectory for a single cell. NS, not significant. Data are mean ± s.e.m. **c**, Normalized nuclear size at the beginning of the time course for proliferatively (prolif.) and formatively (form.)

dividing cells. Two-tailed Mann–Whitney test. Boxes encompass the IQR, centre lines show the median, and whiskers extend to the full range of the data. **d**, Quantified normalized CDT1a–CFP (G1 marker) fluorescence intensity and normalized nuclear size of a complete cell cycle from a light sheet PlaCCI time course. Top, CDT1a–CFP and H3.1–mCHERRY confocal images showing raw signal, and the Otsu threshold mask. Plot and images are representative of 45 cells from independent timecourse experiments with 2 roots. Scale bar, 5 µm. **e–g**, Frequency of dividing cells in unsynchronized roots (**e**), and roots synchronized with hydroxyurea at G1/S (**f**) or oryzalin at G2/M (**g**). Blue, formative; yellow, proliferative. **h**, Percentage of first divisions after dex induction that were formative for roots shown in **e–g**. Unpaired one-sided Student's t-test P is shown. Data are mean ± s.e.m.

were able to find parameter regimes that fit the data reasonably well (Extended Data Fig. 4a,b). However, in many of these cases the model no longer displays bistable properties (Extended Data Fig. 4c).

For the three basic ODE models, the fits were comparable (Fig. 2e–h; Michaelis–Menten: adjusted $R^2 = 0.987$; Hill: adjusted $R^2 = 0.996$; positive feedback: adjusted $R^2 = 0.994$). The best fit Hill coefficient for the Hill model was larger than 1, which can occur due to the existence of positive feedback[30] (Supplementary Methods). This model and the model explicitly incorporating positive feedback visually appeared to capture the rise of SCR better than the Michaelis–Menten model. This finding is consistent with previously described SCR autoregulation[31].

It is not surprising that we were able to fit the data to the Cruz–Ramirez[3] model given the larger number of parameters (which can lead to overfitting). However, three other more simple monostable alternatives with fewer parameters also fit the data well. We conclude that bistability is not required to explain the regulatory relationship between SHR and SCR.

## SHR and SCR act early in the cell cycle

To further investigate the assumption that bistable steady-state levels of SHR and SCR determine whether a cell will divide formatively, we examined SHR and SCR accumulation just prior to division, when the trajectories of both factors have reached high levels. We found variability in SCR levels and did not observe a bimodal distribution corresponding with the fate of the cell (Extended Data Fig. 5a–d). Furthermore, SCR often did not appear to reach a stable point (Extended Data Fig. 5e,f).

Therefore, we hypothesized that a threshold amount of SHR and SCR triggers formative division at an earlier timepoint. To test this,

we determined the accuracy of predicting formative division across a range of SHR and SCR thresholds (Fig. 3a and Supplementary Methods). Optimal thresholds were low relative to the full range of SHR and SCR levels and were able to accurately predict formative division 80% and 73% of the time, respectively. A similar analysis of the SHR confocal data found a maximum prediction accuracy of 87% (Fig. 3a).

To improve these predictions, we considered the possibility that dynamic features of the SHR and SCR trajectories or position in the cell cycle may contribute to the decision to divide formatively. We took a simple machine learning approach (Supplementary Methods) to determine whether we could predict which cells divide using a set of features describing various aspects of the dynamics of the SHR, SCR and nuclear size trajectories (for example, maximum rate, mean SHR and area under the curve; Supplementary Table 1). We used nuclear size as a proxy for position in the cell cycle[11] (Fig. 3b and Supplementary Methods). Our learning model was able to predict whether a cell divides formatively 89% and 92% of the time for the light sheet and confocal data, respectively.

To determine the most predictive features, we assessed the ability of each individual feature to discriminate between formatively dividing and non-dividing cells (Supplementary Methods). In addition to features associated with SHR levels, features relating to nuclear size were significant predictors of formative division (Supplementary Tables 2 and 3), suggesting that threshold levels of SHR might be required during a specific window of the cell cycle for formative division to occur.

To test this hypothesis, we determined the accuracy of predicting formative versus proliferative division for each individual cell cycle based on whether a threshold of SHR or SCR was reached during one of four quarters of the nuclear size range (Supplementary Methods).

Requiring the threshold for SHR or SCR to be met in the first quarter of the nuclear size range resulted in higher accuracies than the predictions using threshold alone (90%, 94% and 89% for the confocal SHR, light sheet SHR and light sheet SCR, respectively) and higher accuracy than requiring the threshold to be met in any of the other three quarters of the nuclear size range (Fig. 3b). In addition, the feature most predictive of formative division over a single cell cycle was the maximum SHR level during the first quarter of the nuclear size range, which accurately predicted 94% and 89% of the confocal and light sheet datasets, respectively (Supplementary Tables 4 and 5). Cells that divided proliferatively had significantly larger nuclei at the beginning of the time course compared with formatively dividing cells (Fig. 3c), suggesting that these cells were already past a critical window of the cell cycle when SHR first reached threshold levels. Using the PlaCCI line[32], which contains a marker for G1, we found that the 25th percentile of nuclear size falls within G1 or up to 1 h after G1 89% ± 1% of the time (mean ± s.e.m.; $n$ = 36 cells from 2 roots) (Fig. 3d, Extended Data Fig. 6a and Supplementary Video 9). This suggests that SHR and SCR are required during G1 or early S phase to trigger a formative division.

To experimentally validate the hypothesis that position in the cell cycle determines sensitivity to SHR, we synchronized the cell cycle throughout the root meristem prior to induction with dex by treating roots containing the inducible SHR construct with hydroxyurea (see Supplementary Methods), which causes cell cycle arrest at the G1/S transition and early S phase[33]. We anticipated that this treatment would result in larger numbers of cells exposed to SHR during the critical early cell cycle window, leading to greater numbers of formatively dividing cells. Consistent with our hypothesis, 94% ± 3% of the first divisions after dex induction ($n$ = 3 roots) in hydroxyurea-treated roots were formative compared to only 52% ± 4% of cells ($n$ = 3 roots) treated with dex alone. Conversely, after synchronization with oryzalin at a later stage of the cell cycle (G2/M) followed by dex treatment, only 20% ± 6% of first divisions after dex treatment were formative (Fig. 3e–h, Extended Data Figs. 6b and 7a–c and Supplementary Video 10).

To understand how these findings inform division of the CEID in wild-type plants, we investigated the dynamics of SHR–GFP and SCR–mKATE2 driven by their native promoters in plants with a wild-type phenotype. We found that levels of SHR in the CEID fluctuated but never appeared entirely absent. SHR and SCR returned to pre-division levels quickly after division of the CEI ($n$ = 6; Extended Data Fig. 8a–g, Supplementary Video 11 and Supplementary Data 3). Thus, it is likely that SHR and SCR are always present early in the cell cycle of the CEID, providing the conditions necessary for formative division.

## Discussion

How developmental regulators control cell division is a central question in developmental biology, with potentially broad applications in understanding basic cell cycle control. Although we cannot exclude the possibility of bistability without a definitive test for hysteresis[34,35], which would be nearly impossible to perform in our system, our data suggest that SHR and SCR are unlikely to regulate formative cell division through a bistable mechanism. Bistability was proposed to explain how and where the decision to divide formatively is made[3]. Our data suggest an alternative mechanism must exist to restrict SHR–SCR-regulated formative divisions to the CEID in wild-type plants. Levels of SHR and/or other coregulators are possible candidates[36,37].

Our finding that low transient levels of SHR and SCR can alter the orientation of the division plane is consistent with previous reports[12,31]. A window of sensitivity to these transcription factors in G1 and early S is consistent with the known role of D-type cyclins, including CYCD6, in phosphorylating RBR during the G1/S transition[3]. RBR-associated kinase activity peaks during the G1/S transition and early S phase[38]. Previous studies have suggested that developmental cues specifying division plane orientation are perceived during G1, much earlier than

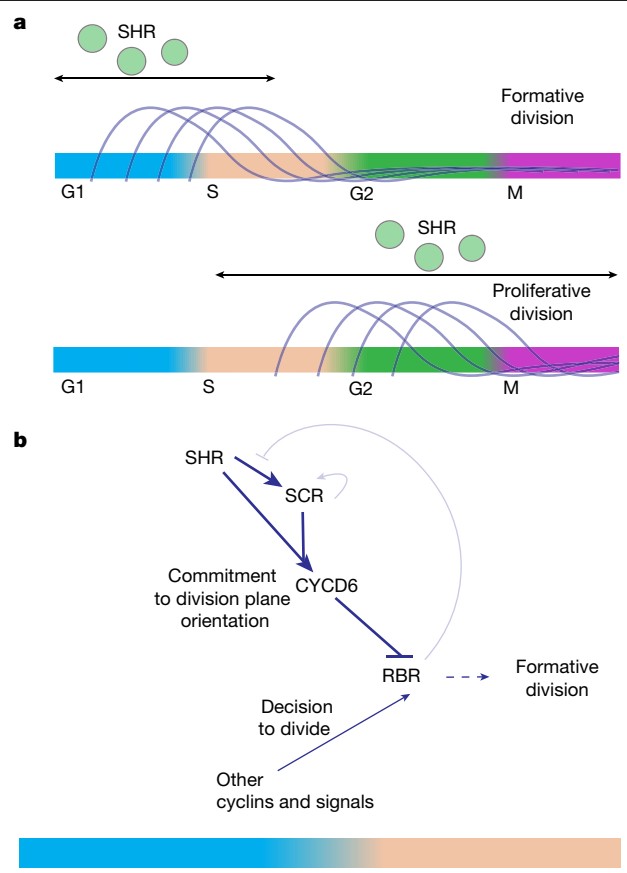

**Fig. 4 | A model for SHR and SCR control of formative division. a**, Threshold levels of SHR and SCR specify formative division only when present during G1 or early S. **b**, The presence of SHR and SCR during G1 and early S activates CYCD6 to specify the orientation of the division plane, whereas other cyclins and developmental cues commit the cell to division. CYCD6 and other cyclins along with their associated kinases phosphorylate RBR, committing the cell to formative division. The two positive feedback loops (SCR autoregulatory loop and RBR release of SCR after phosphorylation by CYCD6) have a smaller role in the decision to divide formatively than previously predicted[3].

the first visible signs of division plane formation in G2[5]. SHR and SCR are transcription factors that need time to activate their targets in the regulation of division plane orientation. Thus, it may seem obvious they would need to act early in the cell cycle. However, transcription and translation occur on the order of minutes[39], whereas the median cell cycle time in roots[40] is approximately 12 h. Given these timescales, SHR and SCR could still function effectively much closer to the time of division plane formation. Understanding the early cellular events regulated by SHR and SCR that lead to the altered division plane is an intriguing avenue for future study.

D-type cyclins activate the RB–E2F bistable switch that commits the cell to DNA synthesis and irreversible progression through mitosis[34]. Notably, however, *SHR* induction cannot initiate formative division outside of the meristem, indicating that SHR is not sufficient to trigger cell division by itself. In addition, CYCD6 is not expressed prior to proliferative divisions in the meristem in our inducible system or in wild-type roots[14] (Extended Data Fig. 2b). These findings suggest a non-canonical role for SHR, SCR and CYCD6 in determining the orientation of the division plane but not initiation and commitment to division. Thus, the presence or absence of SHR early in the cell cycle determines whether the cell will divide proliferatively or formatively, but other cyclins and other developmental cues must be present to initiate cell cycle progression (Fig. 4a,b). The RB–E2F bistable switch

acts to integrate the many signals indicating a cell's readiness to divide[41]. These findings suggest that both the timing and orientation of cell division are determined there.

The CYCLIN D–RBR–E2F pathway is highly conserved between plants and animals, including humans[42,43]. Coordination of axis determination and cell cycle progression by G1/S regulators is important for formative division in metazoans[44–46], and D-type cyclins have been implicated in axis determination in metazoans such as *Caenorhabditis elegans*[44]. Thus, our findings may point to a shared mechanism used to coordinate axis and cell fate determination (patterning) with cell cycle progression (growth) across eukaryotic systems. Perturbation of the CYCLIN D–RBR–E2F pathway is estimated to occur during the development of nearly all cancers[47–51], and defects in division plane orientation and formative division have recently been implicated in the genesis of breast and other cancers[7,8]. Most studies of cell cycle control have used single-cell organisms or cell lines. Future studies using an approach similar to the one described here could reveal mechanisms of cell cycle control that are important during the development of multicellular organisms and suggest opportunities for novel therapeutic interventions in cancer pathogenesis or prevention.

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

# Methods

All methods are included in the Supplementary Information.

## Reporting summary

Further information on research design is available in the Nature Portfolio Reporting Summary linked to this article.

## Data availability

The pre-processed image files are available in the Duke Digital Research Data Repository (https://doi.org/10.7924/r46q2571m; datamanagement@duke.edu). Owing to their large size, original image files are available upon request (please contact researchdatasteward@duke.edu for the first 6 years from publication. To inquire about the availability of this dataset beyond 6 years, please contact caraw97@gmail.com). Complete trajectory data and all metadata needed to run the code are included in the Supplementary material. Source data for figures that were not generated by the code are provided in Excel files. Source data are provided with this paper.

## Code availability

Custom code was central to the conclusions of the paper. The Root-Tracker (for microscope hardware control), image processing and quantification pipeline, and trajectory data analysis pipeline code are available at https://doi.org/10.5281/zenodo.10044880.

**Acknowledgements** We express profound gratitude for the mentorship and unwavering support of Philip Benfey, who recently passed away. He was a guiding light whose visionary thinking shaped the lives and careers of many scientists. We are thankful that he was able to read the final version of this manuscript, and that he could share in the excitement of our findings. We also thank S. DiTalia, L. You, J. Socolar, R. Shahan, R. Sozzani, E. Pierre-Jerome, I. Taylor, T. Nolan, M. Zhu, S. Van Dierdonck, Q. Zhou and O. Szekely for critical reading and discussions of the manuscript; O. Szekely for help with graphics; D. Holland, F. Cutrale and J. Choi for contributing to the light sheet microscope design and construction; and L. Cameron and the Light Microscopy Core Facility at Duke for providing the workstations and support for Imaris image analysis. This work was funded by the US National Institutes of Health (NRSA 5F32GM106690-02 and MIRA 1R35GM131725) to C.M.W. and P.N.B., and by the Howard Hughes Medical Institute to P.N.B. as an Investigator. M.J., S.E.F. and T.V.T. were supported by the Translational Imaging Center, Bridge Institute, University of Southern California.

**Author contributions** C.M.W. and P.N.B. conceived the project and designed the experiments. C.M.W. and H.B. conducted experiments and generated transgenic plants. T.V.T. developed the light sheet imaging platform, with contributions from M.J. and S.E.F. V.P. developed the microscope control code and the image data extraction pipeline. C.M.W., H.B and R.C. performed image analysis. P.S. developed computational tools and performed data analysis of the trajectories. C.M.W. and P.S. interpreted the results and wrote the paper with comments from all authors.

**Competing interests** P.N.B. was the co-founder and Chair of the Scientific Advisory Board of Hi Fidelity Genetics, Inc, a company that works on crop root growth. The other authors declare no competing interests.

**Additional information**
**Correspondence and requests for materials** should be addressed to Cara M. Winter or Pablo Szekely.

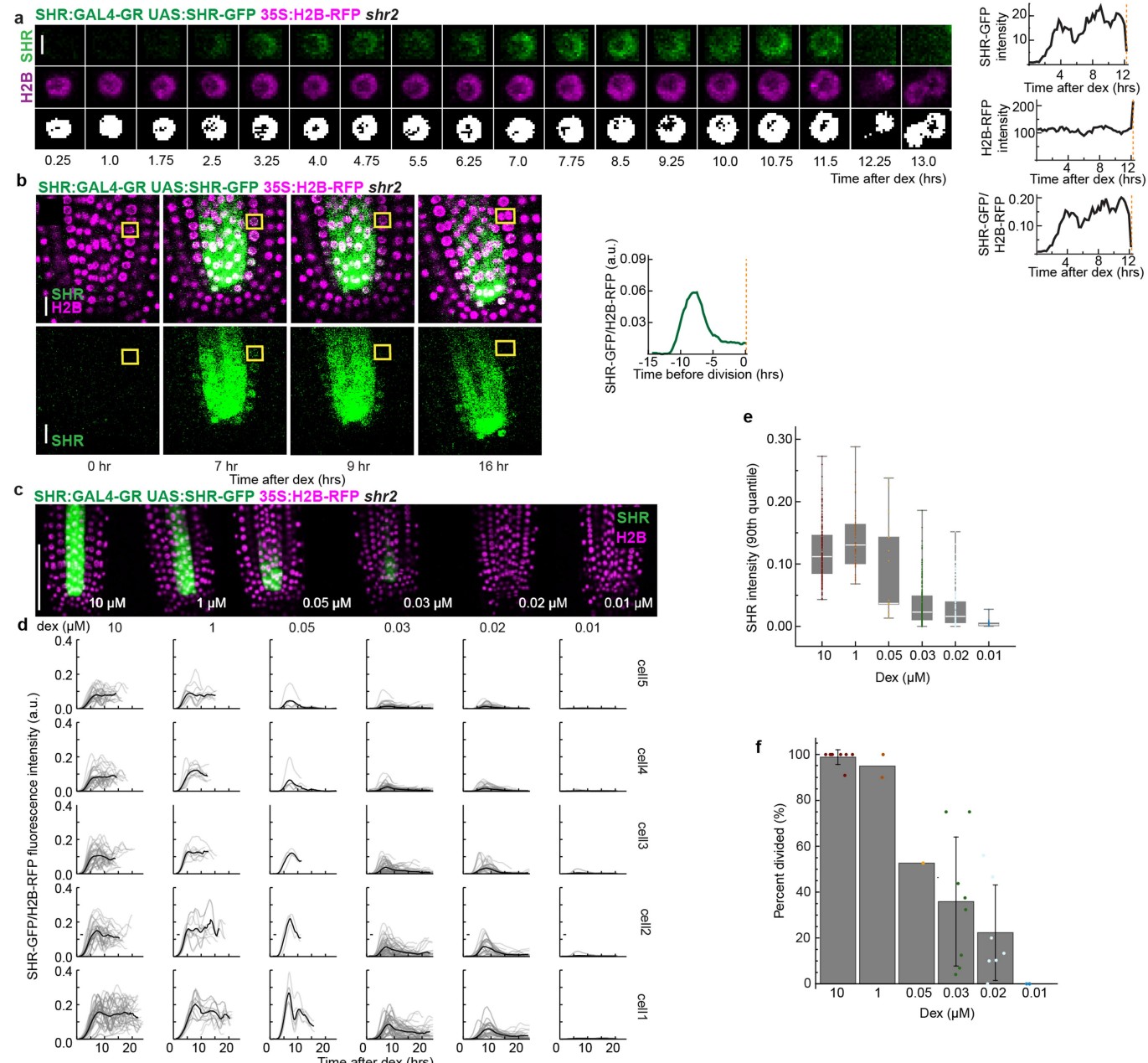

**Extended Data Fig. 1 | Inducible SHR system produced a variety of protein accumulation trajectories and division outcomes. a**, Left, raw confocal images of SHR-GFP (green) and H2B-RFP (magenta), and nuclear mask (white; see Supplementary Methods) of a single cell over time. Every third timepoint is shown. Scale bar, 5 μm. Right, quantification of SHR-GFP (top) and H2B-RFP (middle) signal intensities, and the derived SHR trajectory (bottom). SHR-GFP and H2B-RFP signal intensities were extracted from the region demarcated by the nuclear mask at each timepoint. Images and plots are representative of 935 cells from independent timecourse experiments with 29 roots. **b**, Confocal median longitudinal sections through a root tip treated with low dex (0.02 μM) highlighting another cell that divides proliferatively hours after a transient low peak of SHR-GFP is detected. Quantified SHR trajectory is on the right. Scale bar, 10 μm. **c**, Confocal median longitudinal sections acquired 18 hrs after induction with 10, 1, 0.05, 0.03, 0.02, and 0.01 μM dex. Images are representative of 8, 2, 1, 8, 7 and 3 roots for 10, 1, 0.05, 0.03, 0.02 and 0.01 μM dex, respectively. Scale bar, 50 μm. **d**, SHR trajectories for all cells broken out by dex concentration and cell position. SHR trajectories show a quantitative response to different dex concentrations. Grey lines, raw data. Black lines, smoothed averages. 10 μM, $n$ = 211 cells from 8 roots; 1 μM, $n$ = 63 cells from 2 roots; 0.05 μM, $n$ = 25 cells from 1 root; 0.03 μM, $n$ = 291 cells from 8 roots; 0.02 μM, $n$ = 221 cells from 7 roots; 0.01 μM, 124 cells from 3 roots. **e**, Boxplots of maximum SHR intensity (90th quantile) from all SHR trajectories from all roots treated with different concentrations of dex. Boxes, IQR; centre lines, median, whiskers, full range of the data. **f**, Percentage of cells that divided proliferatively by dex concentration. Data are mean ± s.d. For **e,f**, 10 μM, $n$ = 158 cells from 8 roots; 1 μM, $n$ = 46 cells from 2 roots; 0.05 μM, $n$ = 19 cells from 1 root; 0.03 μM, $n$ = 236 cells from 8 roots; 0.02 μM $n$ = 180 cells from 7 roots; 0.01 μM, $n$ = 104 cells from 3 roots.

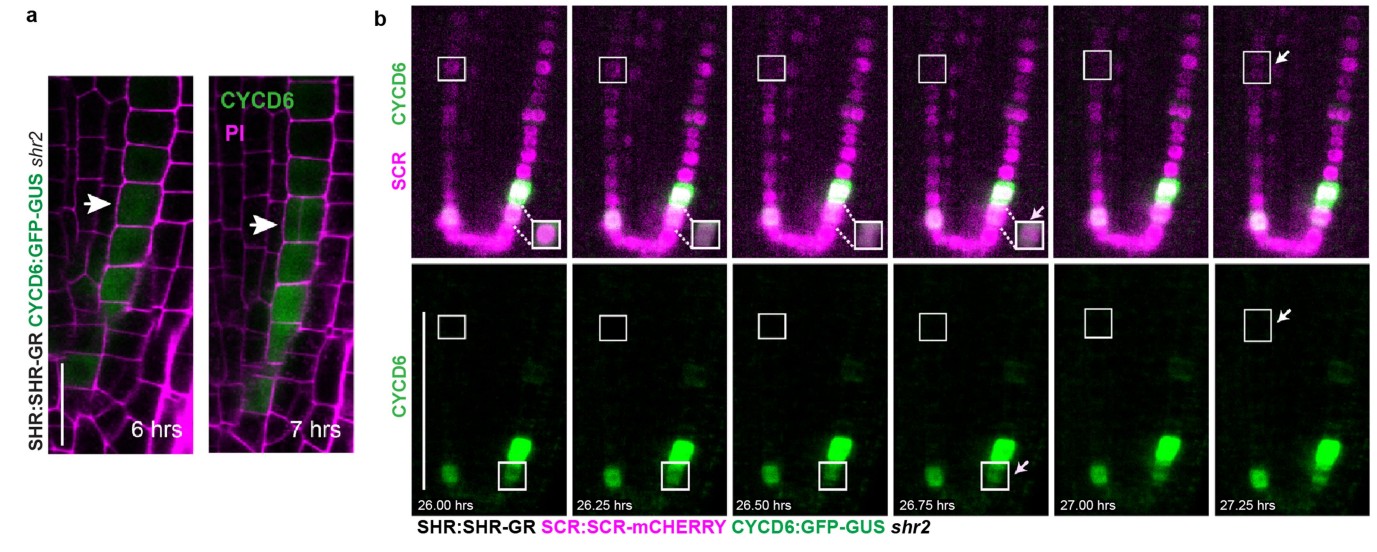

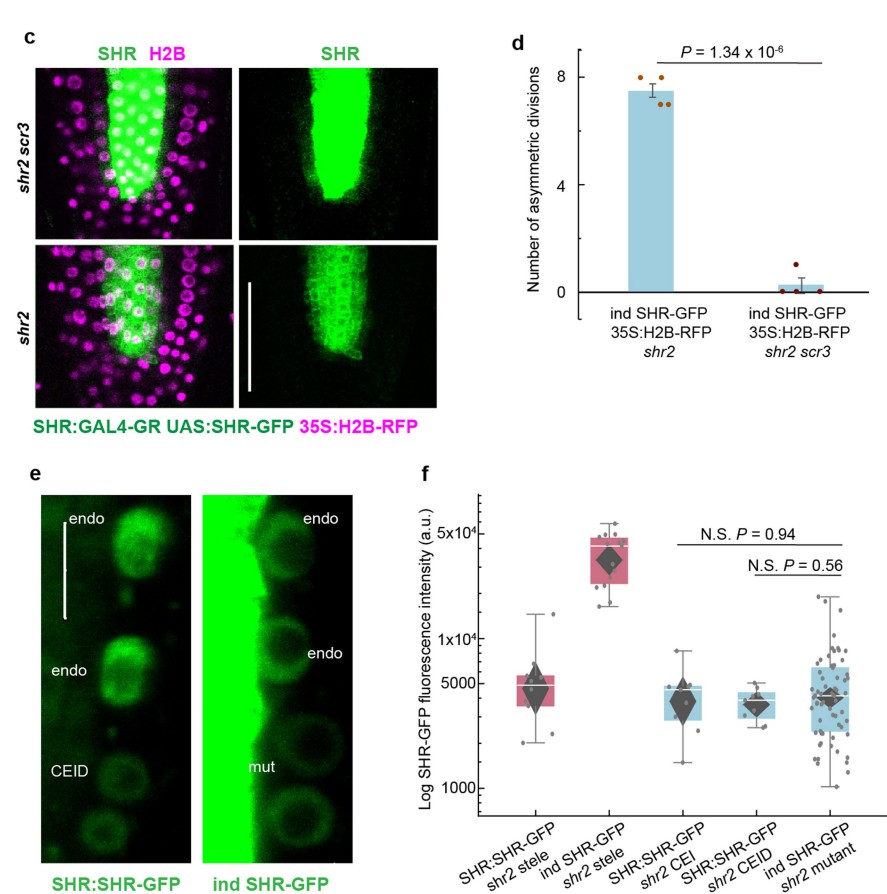

**Extended Data Fig. 2** | See next page for caption.

**Extended Data Fig. 2 | Inducible SHR system controls the SHR-SCR-CYCD6 pathway active in the stem cell niche. a**, Confocal images of a *SHR:SHR-GR CYCD6:GFP shr2* root 6 and 7 h after induction with 10 µM dex. White arrows, cell that divides proliferatively at 7 h. Scale bar, 10 µm. **b**, Confocal images showing *SCR:SCR-mCHERRY* and *CYCD6:GFP-GUS* expression after induction of SHR with 10 µM dex. Maximum pixel intensity for the magenta and green channels is adjusted to enhance visibility of the nucleus in the white box. Inset, image shown with a higher maximum pixel intensity to reduce saturation for that cell. Pink arrow, formatively divided cell; white arrow, proliferatively divided cell. Scale bar, 50 µm. Images in **a**,**b** are representative of independent time courses of 2 roots. **c**, Confocal images of inducible SHR-GFP and H2B-RFP in a *shr2 scr3* (top) or *shr2* (bottom) background after 18-hour 10 µM dex induction. Images are representative of four roots for each mutant line. Scale bar, 50 µm. **d**, Number of formative divisions present in the first five cells of 2 cell files in 6-day old inducible SHR-GFP roots in a *shr2* (*n* = 4 roots) or *shr2 scr3*

(*n* = 4 roots) background after 18 h of dex. Unpaired two-sided Student's t-test *P* is shown. Data are mean ± s.e.m. **e**, Confocal images (green channel only) of a *SHR:SHR-GFP UBQ10:H2B-RFP shr2* root (left) and a fully induced (10 µM dex) *SHR:GAL4-GR UAS:SHR-GFP EN7:H2B-RFP shr2* (right) root 12 h after dex treatment. Images are representative of 5 and 9 roots of the two respective genotypes. Scale bar, 10 µm. **f**, SHR-GFP fluorescence intensity in the stele (*n* = 10 from 5 roots), CEI (*n* = 9 from 5 roots) and CEID (*n* = 9 from 5 roots) of *SHR:SHR-GFP UBQ10:H2B-RFP shr2* plants, and in the stele (*n* = 15 from 8 roots) and mutant cells (*n* = 66 from 9 roots) of *SHR:GAL4-GR UAS:SHR-GFP EN7: H2B-RFP shr2* roots 10–15 h after induction with 10 µM dex. Mean SHR-GFP fluorescence in the *shr2* mutant cells prior to formative division in the inducible SHR line is similar to mean levels of SHR-GFP in the CEI and CEID of *shr2* roots. Mann-Whitney two-sided *P* is shown. N.S., not significant. Boxes, IQR; centre lines, mean, whiskers, full range of the data.

**a**

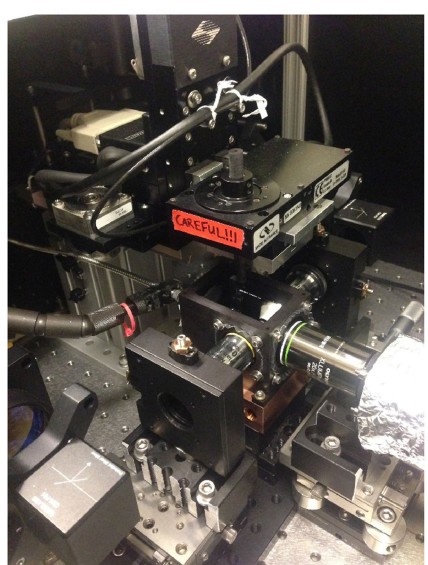

**b**

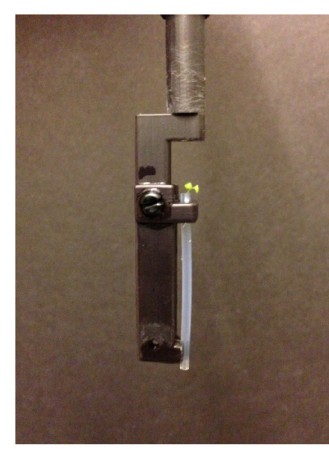

**c**

Sample preparation
and mounting

↓

Root tracking and imaging
(no tracking for confocal)

↓

Image size reduction
Crop, bin, convert to 8-bit

↓

Registration
Register images over time
Register channels to each other

↓

4-D movie generation

↓

Cell tracking
Track individual cells and
quantify gene expression

↓

Background subtraction
and bleedthrough
correction

↓

Smoothing and
normalization

**Extended Data Fig. 3 | Custom light sheet microscope and analysis pipeline.**
**a**, Imaging chamber. **b**, Capillary tube containing growing root mounted onto
custom holder. The holder is lowered into the imaging chamber for imaging.

**c**, Image acquisition and analysis pipeline to produce SHR and SCR trajectories
for confocal and light sheet imaging.

**a**

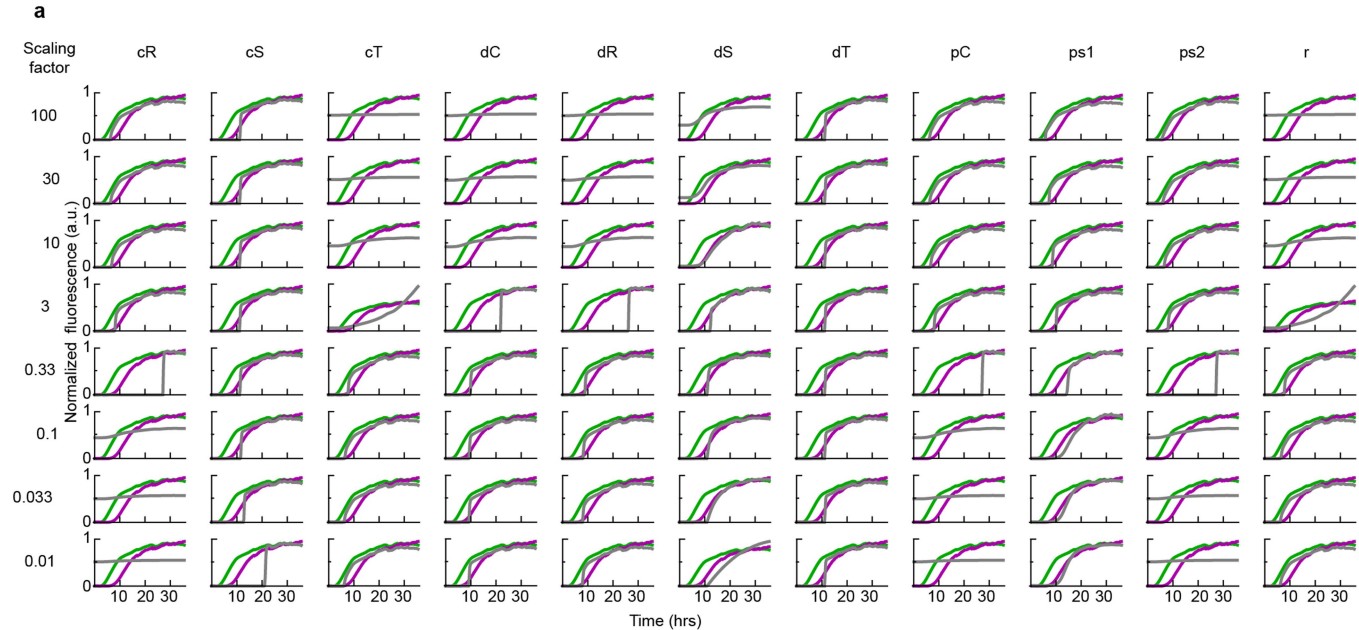

**b**

|        | cR    | cS    | cT    | dC    | dR    | dS    | dT    | pC    | ps1   | ps2   | R     |
|--------|-------|-------|-------|-------|-------|-------|-------|-------|-------|-------|-------|
| 100x   | 0.902 | 0.947 | 0.946 | 0.891 | 0.891 | 0.909 | 0.948 | 0.902 | 0.852 | 0.903 | 0.946 |
| 30x    | 0.901 | 0.948 | 0.954 | 0.916 | 0.917 | 0.967 | 0.948 | 0.901 | 0.856 | 0.902 | 0.954 |
| 10x    | 0.901 | 0.948 | 0.974 | 0.953 | 0.956 | 0.988 | 0.948 | 0.901 | 0.88  | 0.901 | 0.974 |
| 3x     | 0.91  | 0.948 | 0.66  | 0.528 | 0.328 | 0.96  | 0.948 | 0.91  | 0.918 | 0.91  | 0.66  |
| 1/33x  | 0.28  | 0.948 | 0.907 | 0.936 | 0.921 | 0.947 | 0.948 | 0.28  | 0.906 | 0.279 | 0.907 |
| 1/10x  | 0.976 | 0.947 | 0.901 | 0.93  | 0.911 | 0.967 | 0.948 | 0.976 | 0.941 | 0.976 | 0.901 |
| 1/30x  | 0.945 | 0.933 | 0.901 | 0.928 | 0.909 | 0.976 | 0.948 | 0.945 | 0.975 | 0.945 | 0.901 |
| 1/100x | 0.913 | 0.559 | 0.902 | 0.928 | 0.908 | 0.898 | 0.948 | 0.913 | 0.977 | 0.913 | 0.902 |

**c**

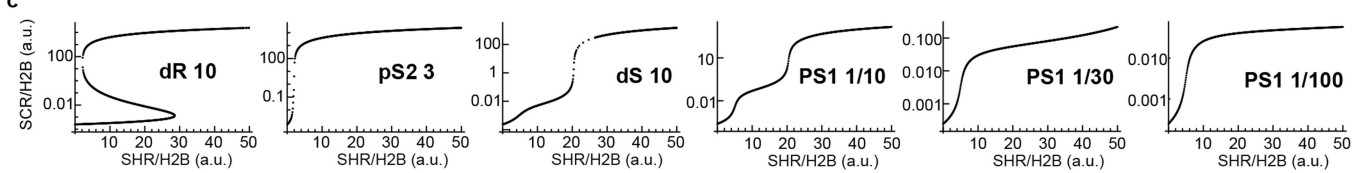

**Extended Data Fig. 4 | Fitting the SCR data to the Cruz-Ramirez model.**
**a**, Mean normalized trajectories of SHR-GFP (green) and SCR-mKATE2
(magenta) ($n$ = 274 cells from 9 roots treated with 40 μM dex; Supplementary
Methods) and predictions of SCR (grey lines) using the Cruz-Ramirez model[3]
(Supplementary Methods) starting from the published Cruz-Ramirez[3]
parameters. Each parameter (columns) was scaled separately by different

values (rows). The resulting SCR curve was then scaled to be comparable with
the measured SCR curve. **b**, The corresponding adjusted $R^2$ for the plots in **a**.
**c**, Example steady state plots for SHR/H2B and SCR/H2B using the Cruz-Ramirez
model (Supplementary Methods) for six of the parameter sets from **a**. The plot
on the left (where the Cruz-Ramirez parameter value for dR was multiplied by 10)
shows bistability, while other parameter sets to the right are monostable.

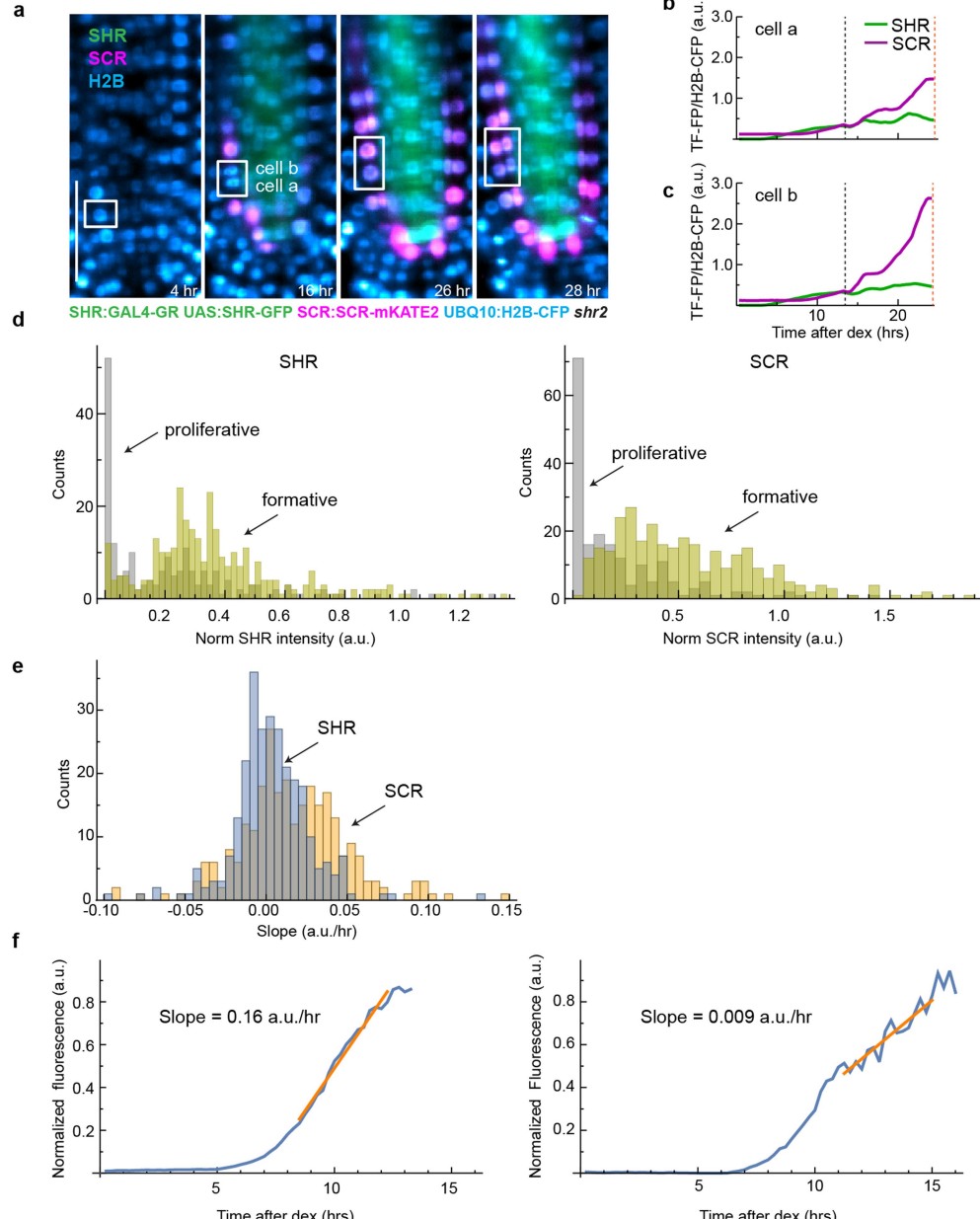

**Extended Data Fig. 5 | SHR and SCR levels at the time of division are not a critical factor in the decision to divide formatively. a**, Median light sheet longitudinal sections of a *SHR:GAL4-GR UAS:SHR-GFP SCR:SCR-mKATE2 UBQ10:H2B-CFP shr2* root treated with 40 μM dex showing two cells from a common progenitor with different levels of SCR just prior to formative division. Images are representative of 16 cell pairs from independent time courses of 10 roots. Scale bar, 50 μm. **b**,**c**, Quantification of transcription factor fluorescent protein (TF-FP) trajectories for the cells in **a**. **d**, Histograms of the average normalized (Supplementary Methods) SHR and SCR levels found during the last five timepoints of all light sheet SHR and SCR full trajectories (including all dex concentrations). Yellow, formatively dividing cells; grey, proliferatively dividing cells. *n* = 500 cells from independent time courses with 14 roots. **e**, Histograms of the slopes for SHR (blue) and SCR (orange) fully induced (40 μM) trajectories. *n* = 274 cells from independent time courses with 9 roots. **f**, Examples of SCR trajectories (blue) and their fitted slopes (orange). For **e** and **f**, slopes were calculated for trajectories between 1 and 5 h prior to division.

**a**

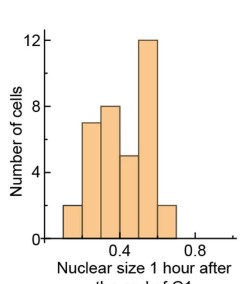

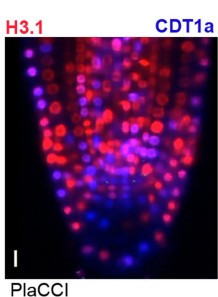

PlaCCI

**b**

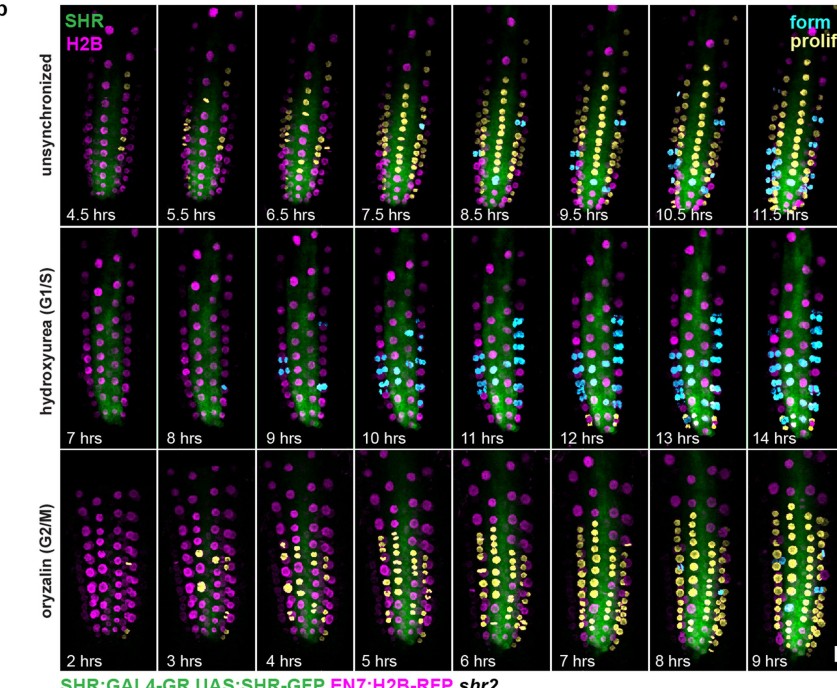

SHR:GAL4-GR UAS:SHR-GFP EN7:H2B-RFP *shr2*

**Extended Data Fig. 6 | SHR levels are interpreted within the context of the cell cycle. a**, Left, histogram of normalized nuclear size one hour after the end of G1 (Supplementary Methods; $n$ = 50 cells from independent time courses with 2 roots). Right, light sheet image of PlaCCI root showing CDT1a-CFP and H3.1-mCHERRY. Scale bar, 10 μm. **b**, Maximum projection confocal images of unsynchronized (top) and hydroxyurea- (middle) or oryzalin- (bottom) synchronized roots induced with 10 μM dex. Timepoints shown include the 8 h during which most cells first divide. Nuclei of cells are pseudo-colored according to the type of first division after dex treatment. Subsequent divisions maintain the pseudo-colour of the first. Yellow, proliferative division; blue, formative division; green, SHR-GFP; magenta, H2B-RFP expressed from the EN7 promoter. Each row shows a time course of a single root out of 3 roots per condition. Scale bar, 10 μm.

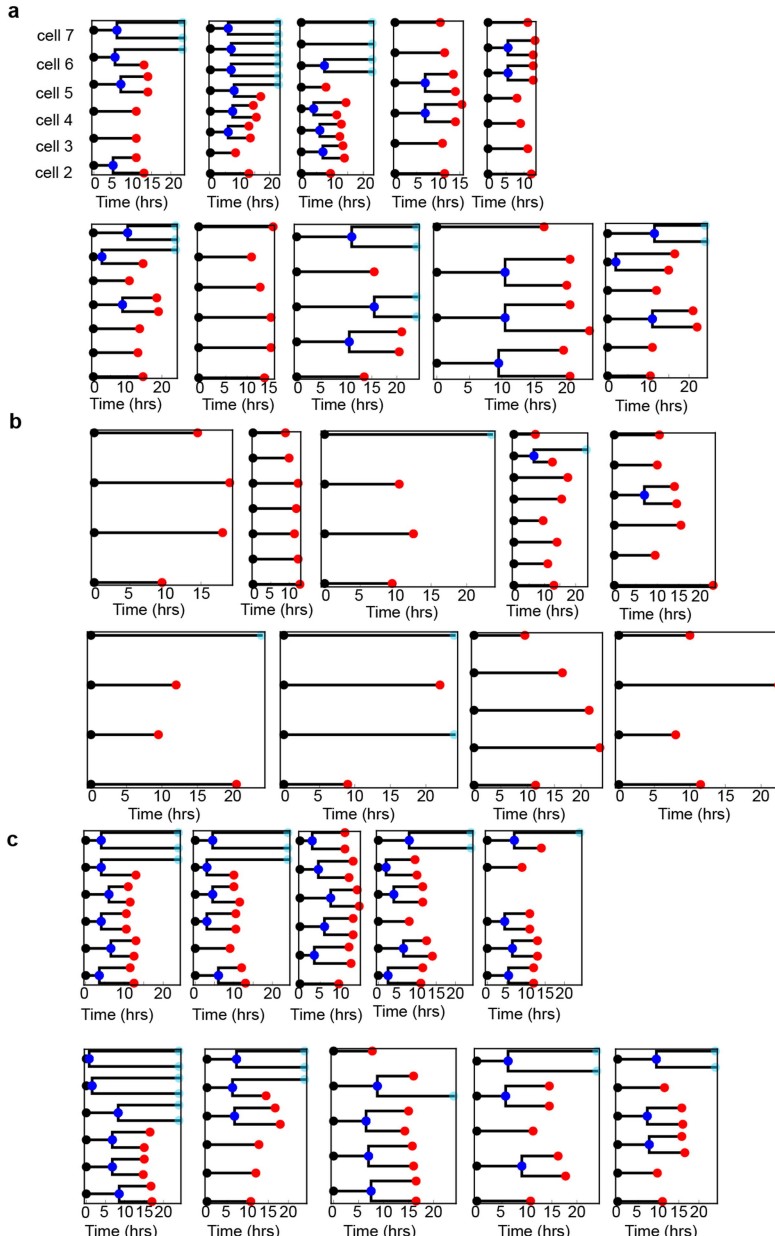

**Extended Data Fig. 7 | Division trajectories of synchronized cells.** Graphical representation showing the timing of proliferative (blue dots) and formative (red dots) divisions for each cell for roots pretreated for 17 h by transfer to plates containing no treatment (**a**), hydroxyurea (**b**), or oryzalin (**c**), followed by transfer to dex for imaging. Each row corresponds to a single root. Data are shown for two roots for each treatment. Each box contains dot plots for cells 2 (bottom) up to 9 (top) from a single cell file. Cells that have not divided proliferatively by the end of the time course have a cyan dot at the last timepoint.

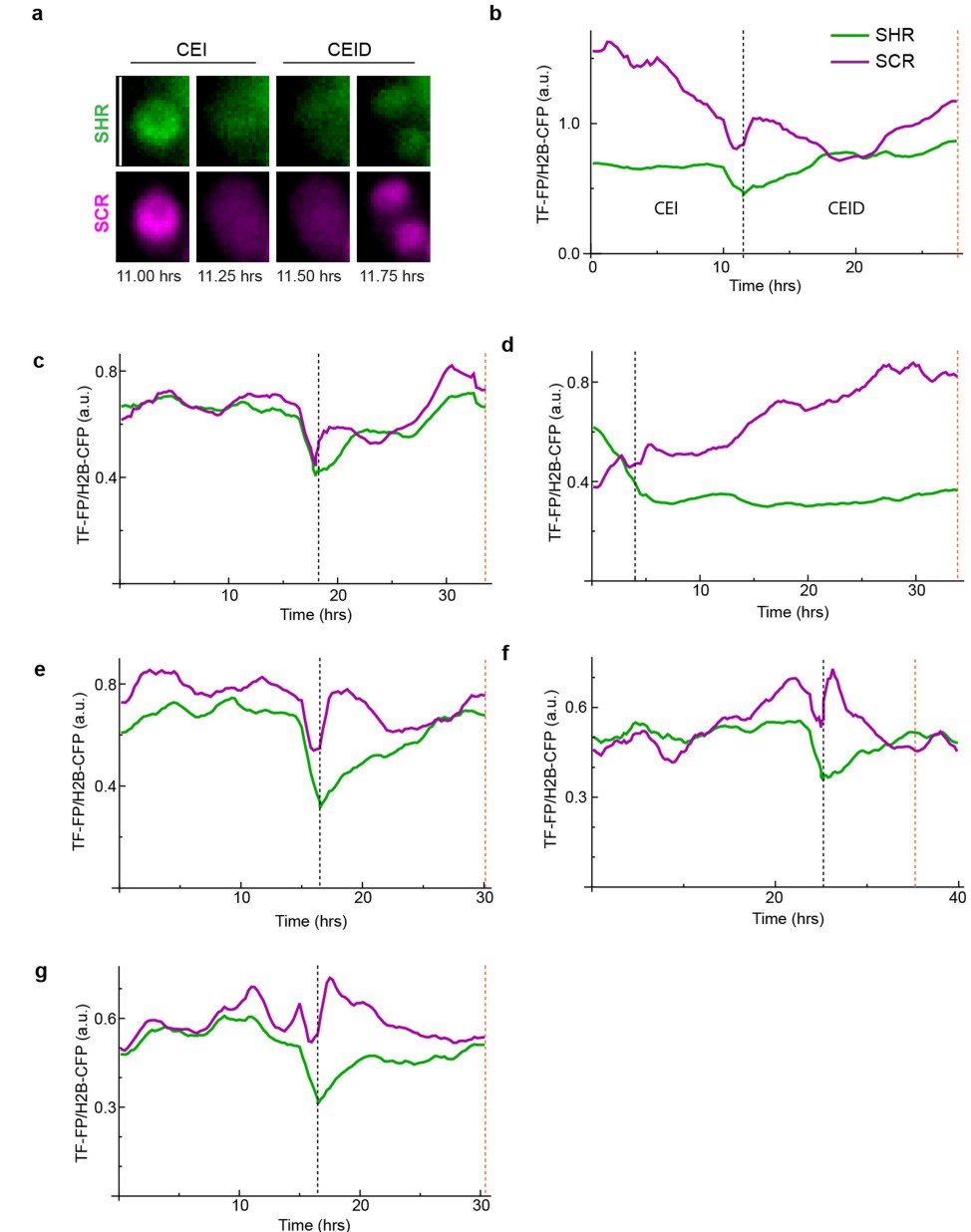

**Extended Data Fig. 8 | SHR and SCR dynamics in CEI and CEID cells. a**, Light sheet images of a CEI cell that divides into a CEI and CEID from a time course of a *SHR:SHR-GFP SCR:SCR-mKATE2 UBQ10:H2B-CFP shr2* root showing SHR-GFP and SCR-mKATE2 fluorescence just before and after CEI division. Images show a single cell representative of six cells from independent timecourse experiments with three roots. Scale bar, 10 μm. **b**, Normalized SHR-GFP and SCR-GFP accumulation in the CEID shown in **a** and its parent CEI during the time course. **c**–**g**, Additional examples of CEID and parent CEI SHR and SCR trajectories. Orange dashed line, formative division; black dashed line, proliferative division. Cells are from three roots.

# Reporting Summary

## Statistics

For all statistical analyses, confirm that the following items are present in the figure legend, table legend, main text, or Methods section.

| n/a | Confirmed | |
|---|---|---|
| ☐ | ☒ | The exact sample size (*n*) for each experimental group/condition, given as a discrete number and unit of measurement |
| ☐ | ☒ | A statement on whether measurements were taken from distinct samples or whether the same sample was measured repeatedly |
| ☐ | ☒ | The statistical test(s) used AND whether they are one- or two-sided<br>*Only common tests should be described solely by name; describe more complex techniques in the Methods section.* |
| ☐ | ☒ | A description of all covariates tested |
| ☐ | ☒ | A description of any assumptions or corrections, such as tests of normality and adjustment for multiple comparisons |
| ☐ | ☒ | A full description of the statistical parameters including central tendency (e.g. means) or other basic estimates (e.g. regression coefficient) AND variation (e.g. standard deviation) or associated estimates of uncertainty (e.g. confidence intervals) |
| ☐ | ☒ | For null hypothesis testing, the test statistic (e.g. *F*, *t*, *r*) with confidence intervals, effect sizes, degrees of freedom and *P* value noted<br>*Give P values as exact values whenever suitable.* |
| ☒ | ☐ | For Bayesian analysis, information on the choice of priors and Markov chain Monte Carlo settings |
| ☒ | ☐ | For hierarchical and complex designs, identification of the appropriate level for tests and full reporting of outcomes |
| ☒ | ☐ | Estimates of effect sizes (e.g. Cohen's *d*, Pearson's *r*), indicating how they were calculated |

*Our web collection on statistics for biologists contains articles on many of the points above.*

## Software and code

Policy information about availability of computer code

| Data collection | For data collection on the confocal microscope, we used Zen 2009 version 6.0.0.303. For data collection on the light sheet microscope we used a custom-written Java application called RootTracker (https://doi.org/10.5281/zenodo.10044880). |
|---|---|
| Data analysis | Data were processed and analyzed using Imaris 9.5.0 and custom scripts written in Python 3.7, Wolfram Mathematica 13.1, R 3.3.3, Fiji 1.50 and 1.52e). Trajectory analysis using Mathematica is available here: https://doi.org/10.5281/zenodo.10044880. |

For manuscripts utilizing custom algorithms or software that are central to the research but not yet described in published literature, software must be made available to editors and reviewers. We strongly encourage code deposition in a community repository (e.g. GitHub). See the Nature Portfolio guidelines for submitting code & software for further information.

## Data

Policy information about availability of data

All manuscripts must include a data availability statement. This statement should provide the following information, where applicable:
- Accession codes, unique identifiers, or web links for publicly available datasets
- A description of any restrictions on data availability
- For clinical datasets or third party data, please ensure that the statement adheres to our policy

The pre-processed image files are available in the Duke Digital Research Data Repository (https://doi.org/10.7924/r46q2571m; datamanagement@duke.edu). Due to their large size, original image files are available upon request (Please contact researchdatasteward@duke.edu for the first 6 years from publication. To inquire

about the availability of this dataset beyond 6 years, please contact caraw97@gmail.com.) Complete trajectory data and all metadata needed to run the code are included in the Supplementary material. Source data for figures that were not generated by the code are provided in Excel files.

# Human research participants

Policy information about studies involving human research participants and Sex and Gender in Research.

| | |
|---|---|
| Reporting on sex and gender | NA |
| Population characteristics | NA |
| Recruitment | NA |
| Ethics oversight | NA |

Note that full information on the approval of the study protocol must also be provided in the manuscript.

# Field-specific reporting

Please select the one below that is the best fit for your research. If you are not sure, read the appropriate sections before making your selection.

☒ Life sciences        ☐ Behavioural & social sciences        ☐ Ecological, evolutionary & environmental sciences

For a reference copy of the document with all sections, see nature.com/documents/nr-reporting-summary-flat.pdf

# Life sciences study design

All studies must disclose on these points even when the disclosure is negative.

| | |
|---|---|
| Sample size | Sample sizes are based on number of roots or number of cells, indicated in the text. The number of cells quantified within one root was determined by how many cells stayed within the meristem for the duration of the timecourse, and were present on the brightest side of the root which was the side closest to the detector (confocal) or camera (light sheet). We made every attempt to replicate every condition with multiple roots, but due to the challenges associated with long-term timecourse imaging and with maintaining a custom-built microscope, this was not always possible. In the two dex conditions where only one root is represented in the data, the data obtained from those follow the trends that are apparent from the other conditions with replicates. |
| Data exclusions | Roots that were not imaged with the same imaging conditions were removed from statistical analyses. Roots that were not imaged successfully for the complete timeperiod (22hrs for the confocal, 45hrs for the light sheet) were excluded. |
| Replication | All conditions were replicated with multiple cells. These came from multiple roots where possible. Due to the challenges of imaging, data from two low dex condition came from single roots. The number of roots and cells used are indicated in the figure legends for each analysis. |
| Randomization | Roots were imaged on different days. All plants were grown under identical growth conditions. All roots of a given genotype were taken from a single transgenic line (single genomic insertion location). |
| Blinding | Investigators performing cell tracking and quantification of gene expression were blind to the dex concentration of the treatment. |

# Reporting for specific materials, systems and methods

We require information from authors about some types of materials, experimental systems and methods used in many studies. Here, indicate whether each material, system or method listed is relevant to your study. If you are not sure if a list item applies to your research, read the appropriate section before selecting a response.

## Materials & experimental systems

| n/a | Involved in the study |
|---|---|
| ☒ | ☐ Antibodies |
| ☒ | ☐ Eukaryotic cell lines |
| ☒ | ☐ Palaeontology and archaeology |
| ☒ | ☐ Animals and other organisms |
| ☒ | ☐ Clinical data |
| ☒ | ☐ Dual use research of concern |

## Methods

| n/a | Involved in the study |
|---|---|
| ☒ | ☐ ChIP-seq |
| ☒ | ☐ Flow cytometry |
| ☒ | ☐ MRI-based neuroimaging |

