## [Peer Review File · Nature]

Manuscript Title: SHR and SCR coordinate root patterning and growth early in the cell cycle

Editorial Notes:

Redactions – unpublished data

Reviewer Comments & Author Rebuttals

Reviewer Reports on the Initial Version:

Referees' comments:

Referee #2 (Remarks to the Author):

The manuscript by Winter et al. describes an experimental study of the dynamics of two transcription factors, SHR and SCR, which control asymmetric cell divisions in the stem cell niche of Arabidopsis roots. The manuscript uses an inducible expression system, where exogenous SHR can be expressed upon induction with dexamethasone, in a *shr2* mutant background. Overexpression of exogenous SHR rescues asymmetric cell divisions in this mutant. The authors use confocal and light sheet fluorescence imaging of fluorescent SHR and SCR protein fusions to track the expression dynamics of the two transcription factors, in relation to asymmetric cell divisions. The authors argue that their data seem to be inconsistent with a previous model proposing a bi-stable SHR-SCR switch (due to two underlying nested feedback loops). Rather their real-time observations indicate that SHR (and to lesser effect, SCR) expression levels need to reach a certain threshold early in the cell cycle (judged by cell size). The authors speculate that these threshold SHR and SCR levels might then trigger downstream bi-stable switches of cell cycle regulators, to determine asymmetric vs. symmetric cell division decisions.

The manuscript could be better written to make certain points more clear and enable readers that are not familiar with cell fate specification literature in plant roots to follow the arguments that the authors are trying to make. Also, I am not sure how strong some of the conclusions can be based on the data presented.

A major limitation of the manuscript is that the exact predictions of the previous work by (Cruz-Ramirez et al., Cell 2012 <https://doi.org/10.1016/j.cell.2012.07.017>) describing the bi-stable switch model are not clearly articulated. It would be very useful to present some specific experimental predictions based on that model (as well as state the assumptions and their validity for the range of parameters and that are applied), and then present a side-by-side comparison of those predictions and the experimental observations. At present, it is not clear if the results in the present paper truly contradict the bi-stable model, especially since the authors do not directly test bistability in the system (e.g. presence of hysteresis or a bimodal distribution of an output for the same input conditions).

It is not clear if the experimental system that the authors study here captures most of the physiology of the endogenous SHR and SCR. Also, the previous work that posited the bi-stable switch model did not consider inducible overexpression of exogenous SHR, which takes the system out of its natural spatiotemporal dynamic range. How physiologically relevant is this system compared to the endogenous expression of SHR? Is the dex induction capturing any of the endogenous SHR spatiotemporal expression dynamics, and does endogenous SHR also need to be expressed early during the cell cycle just before the asymmetric cell divisions? Or is endogenous SHR already expressed and has reached the presumed threshold levels by the start of this cell cycle? How does

the exogenous overexpressed SHR levels compare to the endogenous SHR levels?

Since the authors use an inducible overexpression system, and both SHR and SCR are transcription factors that need to modulate expression of target genes (some of them producing proteins that regulate cell cycle), it is not clear from the text if it is not kind of obvious that SHR and SCR would need to be expressed early in the cell cycle for their activities (which involve new protein synthesis of the downstream targets) to have any significant influence on cell-cycle progression and outcome of cell division.

The predictive statistical analysis of the data is nice, but it does not prove the requirement of certain SHR and/or SCR levels at a specific stage of the cell cycle. Could the overall relationship between SHR/SCR levels and asymmetric vs. symmetric division (or asymmetric division vs. non-division) decisions merely reflect the limitations in the speed of inducible expression of the overexpressed SHR protein? Again, it is not explained whether this artificial scheme recapitulate any of the endogenous SHR spatio-temporal expression patterns.

Finally, although the bistable switch model might not fit the data from the inducible SHR overexpression in the *shr2* background, the authors do not perform any additional experiments that can further elucidate new mechanisms – most of the discussion in the paper is speculative. I am not sure then if the general readership in Nature is thus left kind of unsatisfied.

Additional comments:

Kinetics of gene expression measured using fluorescent protein reporters might need to be corrected for the maturation time of the fluorescent proteins. Experiments in E Coli show temperature dependent maturation rates of up to several hours, with an almost 2-fold increase when temperature drops from 37degC to 32 degC (Balleza, E., Kim, J. & Cluzel, P. Systematic characterization of maturation time of fluorescent proteins in living cells. *Nat Methods* 15, 47–51 (2018). <https://doi.org/10.1038/nmeth.4509>). I wonder what GFP, mKate and RFP maturation times are at the experimental temperature of 21 degC used for the root imaging experiments in the manuscript. This might be important, since some of the results rely on accurate quantification of FP expression dynamics (e.g. Fig 2f-h).

Pg 10. “The lower prediction accuracy obtained using SCR levels suggests that its role in asymmetric division is secondary to that of SHR.” – It is not clear if the slight decrease in predictive power for SCR vs. SHR is biologically significant, or there are technical reasons for this different (e.g. use of mKate vs. GFP, different SNR in imaging the two channels, etc.)

Pg 12-13 “To understand how our findings inform division of the CEID in wild-type plants ...” – this paragraph presents a single observation of one cell division in one experiment! It is not clear if it is appropriate to draw any conclusions from just n=1 observation, and if a more controlled experiment and appropriate statistical analysis should be performed. Otherwise this paragraph and related discussions should be removed.

Some of the experimental parameters are not clear. E.g., Fig 1e-f, Suppl Videos 5&6 – what is the concentration of dex used? The text only specifies “low dex” induction.

Similarly, I could not find in the text the dex induction levels for the data used to construct the predictive models. Are these analyses mostly relying on the “fully induced” 10 uM dex conditions or on the lower (0.02-0.05 uM dex) induction conditions?

The manuscript could be improved by better annotation of some of the figures and, especially, the videos.

Perhaps the authors could better explain how the image acquisition and analysis pipeline in Ext Data Fig 3c relates to the corresponding Methods sub-sections.

Ext Data Fig 4 – x axes needs labels.

Referee #3 (Remarks to the Author):

The article entitled “Patterning and growth are coordinated early in the cell cycle” aims to provide a mechanistic understanding of how two transcription factors contribute to the decision-making between formative and proliferative division in a cell cycle-dependent manner.

The authors used a custom-made light-sheet imaging system to perform quantitative 4D and 3D imaging in vivo combined with mathematical models based on simple ODE.

Comments

The title is too generic, mainly because the manuscript focuses on only two transcription factors; a specific title would be more appropriate for this manuscript.

The authors first report the function of SHR in controlling asymmetric cell division through imaging roots expressing an inducible SHR line in vivo using a light sheet confocal microscope that allowed extended live imaging without bleaching or phototoxicity. While the imaging quality is superb, the outcome is not novel as the function of SHR using this inducible SHR line is well established in the host laboratory, yet, they were useful for the modeling section.

The authors based their analysis on the inducible SHR (SHR:GAL4-GR UAS:SHR-GFP). However, while looking at the confocal images, although the line complemented the *shr* mutants, the expression level in the endodermis was very low when compared to images in previous studies from the Benfey’s laboratory (<https://www.hhmi.org/content/benfey-research-abstract-slideshow>), Nakajima et al., 2001; Cui et al., 2007, etc.). Since the whole manuscript is based on imaging and quantifying SHR levels, the differences in some conclusions with the previously published bistable model might be caused by the differences in SHR levels in the endodermis between the lines used in the two different studies. It is well plausible that the conclusions from this manuscript are valid but performing similar experiments with another inducible line where the SHR promoter could be fused for instance, to an estrogen receptor-based transactivator XVE to drive the SHR protein and then use this line to monitor the SHR dynamics and assess whether the same results are obtained. That would strengthen challenging the previously published model.

Furthermore, the work published by Heidstra et al. in 2004 (Mosaic analyses using marked activation and deletion clones dissect Arabidopsis SCARECROW action in asymmetric cell division) provides excellent tools to study both SHR and SCR roles in controlling the divisions and assess whether these are symmetric or asymmetric. Combining the SCR activation and deletion clones with the inducible SHR in one plant and taking advantage of the light sheet technology and the modeling will provide more substantial evidence and robust data to support the observations obtained in this manuscript.

The authors also provide a new model for SHR and SCR to control asymmetric cell division by regulating CYCD6 while the symmetric division is through other cyclins. They state that SHR and SCR

would activate CYCD6 during the G1 phase of the cell cycle. However, these observations were based on treatment with the cell cycle inhibitor HU, and there was no genetic data to support these conclusions. One way to validate the outcome of this study would be to do these experiments using the CYCD6 marker and show that it is activated only during asymmetric cell division. The authors should also use other cell cycle inhibitors (G2-M phase) to prove that the action is G1-S specific.

Referee #4 (Remarks to the Author):

In this work authors address the regulation and coordination between formative (asymmetric) and proliferative (symmetric) cell divisions, a key question in developmental biology. They analyze in detail these two types of cell divisions in the stem cell niche of the Arabidopsis root using light sheet and confocal microscopy in an inducible system of SHR, which together with SCR, are involved in asymmetric cell division (ACD). Their main claim is that the existing bistable model is challenged. Indeed they obtained results supporting the conclusion that this model does not apply and instead, low and transient levels of SHR are required for ACD. They conclude this based on detailed measurements of SHR kinetics. They also propose that expression of SHR early in the cell cycle determines the type of cell division.

The work is technically challenging and authors master the experimental strategies used. The main claims of this study, namely, the nature of SHR as a driver and that low and transient levels of SHR are sufficient to determine ACD, are supported by the data provided, although some loose ends remain. A major point not sufficiently addressed (and discussed) is whether the known role of CYCD6;1, and other components of the pathway in CEID division, applies to other cells developing ACD. Also, why does an ACD is followed by a symmetric division and not by new ACDs, without restrictions (since the inducible strategy is producing SHR in a continuous manner)?

A general comment is that it is not easy to follow and find all the necessary information in the figure legends. In Fig. 1b, the focus is on 5 distal cells. Later in the text, the term "all cells" is frequently mentioned. Do authors refer to "all these five cells" or to "all cells in the meristem file", or else?

Specific comments

1. Fig. 1d. This figure is not sufficiently described in the text. This panel should include the dividing cell at the end (12h?). There is an apparent decrease of SHR amount at ~6h. Is this observed in other cells analyzed? Does this have a relevance for ACD? In fact, the accumulation pattern seems to be different in different cells: maximum levels reached towards the end of the cell cycle (cell #5), a peak early in the cell cycle and then steady levels (cell #2), a peak and then a decrease (cells #1 and #4). Please comment. SHR sharply disappear coinciding with mitosis. Is this the results of targeted proteolysis and/or dilution out of chromatin?

2. Fig. 3f. The prediction accuracy is close to >90% for smaller cells. However, it is also quite high (~50%) for cells with larger sizes. Does this mean that the proposed relevance of SHR level early in the cell cycle is important but that high levels later in the cell cycle also lead to ACD?

3. Ext. Data Fig 2i. Cells completing a cell cycle in such a short time, both asymmetric and symmetric cell divisions, is unprecedented in the literature. This is also difficult to reconcile with results shown in Fig 3k, in the absence of hydroxyurea, when a high fraction of cells respond to SHR dividing asymmetrically (and therefore, responding early in the cell cycle, as proposed here). Such high fraction is suggestive of a longer (early) cell cycle, otherwise the window of opportunity to respond to SHR would be much smaller.

4. Induction of CYCD6;1 expression is a requirement to determine ACD in the stem cell. Authors do not show whether this is also the case in the cells upon SHR expression is induced. Does ACD in cells (other than the stem cell) depends on CYCD6;1? Is the auxin balance relevant for the ACDs in these cells?

5. Fig. 4. Showing direct measurements of SHR accumulation in newly-formed cells after a symmetric cell division would be a strong support to the main claim. This will also provide a more precise identification of when SHR level is critical early in the cell cycle.

6. What is the expression pattern of cell fate genes after Dex treatment?

Author Rebuttals to Initial Comments:

Response to Referees

Comments from Referee #2:

Comment 1:

“The manuscript by Winter et al. describes an experimental study of the dynamics of two transcription factors, SHR and SCR, which control asymmetric cell divisions in the stem cell niche of Arabidopsis roots. The manuscript uses an inducible expression system, where exogenous SHR can be expressed upon induction with dexamethasone, in a shr2 mutant background. Overexpression of exogenous SHR rescues asymmetric cell divisions in this mutant. The authors use confocal and light sheet fluorescence imaging of fluorescent SHR and SCR proteins fusions to track the expression dynamics of the two transcription factors, in relation to asymmetric cell divisions. The authors argue that their data seem to be inconsistent with a previous model proposing a bi-stable SHR-SCR switch (due to two underlying nested feedback loops). Rather their real-time observations indicate that SHR (and to lesser effect, SCR) expression levels need to reach a certain threshold early in the cell cycle (judged by cell size). The authors speculate that these threshold SHR and SCR levels might then trigger downstream bi-stable switches of cell cycle regulators, to determine asymmetric vs. symmetric cell division decisions.

The manuscript could be better written to make certain points more clear and enable readers that are not familiar with cell fate specification literature in plant roots to follow the arguments that the authors are trying to make.”

Response: To make the manuscript more accessible to a broad audience we have changed both the writing and the figures in many ways that we think have improved its clarity. Some of the specific changes we made are listed here:

- We moved the diagram showing the root morphology (previously located in Extended Data Figure 1) to the main Figure 1a. and modified it to illustrate SHR expression dynamics in wild type and our inducible system.
- We added a diagram of the regulatory network modeled on the Cruz-Ramirez 2012 paper (Figure 1b).
- We removed Figure 1g-l, and the corresponding text in the manuscript. This was a confusing point, and the conclusions were weaker after correction for maturation time.
- We have clarified that using the SHR induction system was not intended to capture the endogenous SHR kinetics, but instead was utilized to directly modulate SHR dynamics in order to probe the aspects of those dynamics that are required for SHR function (rescuing asymmetric division). This was not clear in the original manuscript.

Comment 2. *“Also, I am not sure how strong some of the conclusions can be based on the data presented.”*

Response: We have added several new experiments and analyses that we think have strengthened the manuscript and its conclusions. These include:

1. **Further testing for the presence of bistability.** We completed the following analyses and experiments:
 - We analyzed the endpoint distributions of SHR and SCR at the time of division. We found no evidence for a bimodal distribution of levels that correlates with the fate of the cell (asymmetric or

symmetric division) (see Response to Comment 4). These data are now included as Extended Data Figure 6a.

- We analyzed the slopes of the curves for SHR and SCR at the end of the time courses, showing that in many cases SCR does not appear to reach equilibrium, an underlying assumption of models of bistability (see Response to Comment 4). These data are now included as Extended Data Figure 6b and c.
 - We also included the specific predictions of the Cruz-Ramirez model for comparison to our observed SHR and SCR data, and for comparison to the predictions of the monostable alternative models that we tested (see Response to Comment 3). These predictions are now included in Figure 2e and Extended Data Figure 3e and f.
2. **Better definition of the relationship between nuclear size and cell cycle stage.** We imaged 2 roots of the PlaCCI line (Desvoyes *et al.*, 2020), which contains a marker for G1 (CDT1a). This marker is expressed during the G1 phase of the cell cycle and is rapidly degraded at the onset of S. The PlaCCI line also contains an H3.1 marker that is expressed throughout the cell cycle in the proliferating cells found at the root tip. We used this marker to determine the size of the nucleus. Through analysis of light sheet images we determined that the 25th percentile of the nuclear size range falls either within G1 or up to 1 hour after G1 89% of the time. These data add support to our finding that SHR is required early in the cell cycle and specify that timeframe likely includes G1 or early S phase. We have incorporated these data into Figures 3g and h.
 3. **Further experimental confirmation of our in-silico-derived hypothesis.** In the previously submitted manuscript we included experimental confirmation of our hypothesis that SHR is required early in the cell cycle by treating roots with hydroxyurea prior to induction with dex. By inducing SHR in these roots containing a greater number of cells at an early cell cycle phase we observed a greater number of asymmetric divisions. In this revision, we have added another cell cycle inhibitor, specific to G2/M. Complementary to our previous findings, we found that inducing SHR in roots containing a greater percentage of cells at a later cell cycle phase resulted in fewer asymmetric divisions. These results provide strong experimental support for our finding that SHR is required early in the cell cycle to specify asymmetric division. We also changed the presentation of these data in the figure to make them more visible. These data are now included in Figure 3i-m.
 4. **Additional imaging of SHR and SCR dynamics in wild-type plants.** We were able to capture 5 additional time courses of SHR and SCR expression in their native context that corroborated the results we obtained from the first time course presented in the submitted manuscript. We imaged fluorescently labelled SHR, SCR, and a nuclei marker in cortex/endodermal initial daughter (CEID) cells in plants with a wild-type phenotype. These time courses span the time from creation of the CEID up through its asymmetric division and show that there is not a consistent pattern of SHR and SCR expression. However, SHR and SCR are visibly present immediately after division of the CEID, which is sufficient in our model to alter the division plane of the cell, resulting in asymmetric division. The data for these time courses is provided in Supplementary Data File 3.
 5. **Further validation that use of our inducible system to understand the behavior of endogenous SHR is sound.** In this revision, we show that CYCD6 is expressed prior to asymmetric divisions in our inducible system, but not prior to symmetric divisions. This parallels the behavior of CYCD6 in WT plants, where it is expressed prior to asymmetric division of the CEID, but not prior to symmetric divisions in the meristem. We also show that levels of SHR expression prior to asymmetric division in

our inducible line are comparable to the levels of SHR expression in the CEID in wild-type plants. These data are included in Extended Data Figure 2d and e, and Extended Data Figure 8.

6. Confirmation that maturation kinetics of fluorescent proteins do not affect our main conclusions.

We corrected the trajectory data for maturation time using an ODE model (see below) and reran our analysis pipeline. The results confirmed all of our main conclusions and are included in Extended Data Figure 9. One conclusion (previously Figure 1g – i) was not as strong after the correction, so we removed it from the figure and text.

Comment 3. *“A major limitation of the manuscript is that the exact predictions of the previous work by (Cruz-Ramirez et al., Cell 2012 <https://doi.org/10.1016/j.cell.2012.07.017>) describing the bi-stable switch model are not clearly articulated. It would be very useful to present some specific experimental predictions based on that model (as well as state the assumptions and their validity for the range of parameters and that are applied), and then present a side-by-side comparison of those predictions and the experimental observations.”*

Response: This is an important point. We deliberated over whether to include the specific quantitative predictions of the Cruz-Ramirez model in the original manuscript. We chose to leave them out based on the advice of a mathematical modeler here at Duke who read a draft that contained the predictions. Given the complexity of the Cruz-Ramirez model (17 parameters and 6 differential equations), it would be considered overfitting if we were to attempt to fit the model to the observed mean SCR trajectory using all possible combinations and covering the full range of all the parameters. Thus, it is impossible to say exhaustively that the Cruz-Ramirez model does not fit the data. The Cruz-Ramirez model falls within a wider family of possible bistable models, and we believe the evidence we provide challenges the presence of bistability, in general, in this system. However, we do make claims that our observations are not consistent with the predictions of the Cruz-Ramirez model, and we agree that showing the predictions clarifies the qualitative observations that we make.

To attempt to thread this needle, we now show in Figure 2e the predictions of the Cruz-Ramirez model using their published parameters. We have also included in Extended Data Figures 5a and b a grid of a series of predictions of the model, where we varied each parameter individually by 5 orders of magnitude (2 above the published parameter, and 2 below it). Using this approach, it is possible to find a set of parameter values that results in a predicted SCR trajectory that coincides with the observed SCR data reasonably well. However, we note in the text that in many cases the model no longer displays bistable properties (shown in Extended Data Figure 5c).

We showed in the submitted and revised manuscripts that it is possible to fit the data using simpler models with many fewer parameters (MM: 1 equation, 3 parameters; Figure 2f; Hill: 1 equation, 4 parameters; Figure 2g; Linear feedback: 1 equation, 6 parameters; Figure 2h). Based on this finding, we make the point that the model of bistability is not needed to explain the data that we observe. We invoke the principle of Occam’s razor to suggest that the simplest explanation is usually the best one. And we present evidence that the data we observed are inconsistent with qualitative predictions of the model, namely that relatively high steady-state levels of SHR and SCR are required to trigger asymmetric division. Ultimately, we offer a new model as an alternative in which relatively low levels of SHR and SCR present early in the cell cycle trigger asymmetric division.

We were careful with our wording in the submitted manuscript to state that we cannot disprove the model of bistability without a test for hysteresis. We did attempt to do this but were unsuccessful due to

several technical challenges (see below). In this revision, we found one place where we could be even more conservative in our wording and have changed the text accordingly:

Abstract:

Removed: Nuclear SHR kinetics do not follow predictions of the bistable model,

Added: "We show that SHR and SCR kinetics do not align with the expected behavior of a bistable system."

Comment 4. *"At present, it is not clear if the results in the present paper truly contradict the bi-stable model, especially since the authors do not directly test bistability in the system (e.g. presence of hysteresis or a bimodal distribution of an output for the same input conditions)."*

We agree that the ultimate test of a bistable model would be a hysteresis test. We attempted to do this test by growing roots on either no dex or 10 μM dex and transferring them after 5 days to a range of different concentrations of dex (0, 0.01, 0.02, 0.03, 0.04, 0.05, 0.1, 1.0, and 10 μM). We included hydroxyurea in these experiments so that we could measure equilibrium levels of SHR without divisions. In roots transferred from 10 μM to 0 μM dex, SHR-GFP remained in the central tissues of the root, even after 3 days on no dex. The reasons for this perdurance are not clear, however this source of SHR confounds the analysis of hysteresis in the *shr* mutant tissue layer because any residual SHR after reducing the dex concentration could potentially be derived from a bistable network in the mutant layer cells OR from the residual pool of SHR-GFP in the stele.

In this revision, we were able to test further for the presence of bistability by looking for a bimodal distribution of SHR and SCR expression levels. To do this, we analyzed the levels of SHR and SCR just prior to division (average of the last 5 timepoints) for all cells and induction levels. We found that, for SCR, the distribution is unimodal with an accumulation of cells with no detectable SCR levels. The SHR distribution seems to be bimodal. However, the distributions for both SHR and SCR are not correlated with the fates of the cells, as the overlap between asymmetric and symmetrically dividing cells is substantial. These distributions are now included in Extended Data Figure 6a.

An underlying assumption of models of bistability is that the regulators operate at equilibrium. Our observation that transient levels of SHR can trigger asymmetric division suggests that SHR does not operate at a "locked" equilibrium state. In this revision, we show that SCR also does not appear to operate at equilibrium. We analyzed the 15 timepoints up to an hour prior to asymmetric division in fully induced roots and found that the average slope of the SCR trajectories is significantly larger than 0 (mean slope = 0.015 SCR a.u./hr, $P = 5 * 10^{-11}$, Student's T-test). These results are included in Extended Data Figure 6b.

We are careful in the manuscript not to say that we are disproving the bistable model. We state in the Discussion that we cannot exclude the possibility of bistability without a test for hysteresis. Instead, we present results that are inconsistent with this model (e.g. transient low levels of SHR can trigger asymmetric division), show the model is not necessary to explain SHR regulation of SCR, and provide a new model that better explains the data that we observe.

Comment 5. *"It is not clear if the experimental system that the authors study here captures most of the physiology of the endogenous SHR and SCR. Also, the previous work that posited the bi-stable switch model did not consider inducible overexpression of exogenous SHR, which takes the system out of its natural spatiotemporal dynamic range. How physiologically relevant is this system compared to the*

endogenous expression of SHR? Is the dex induction capturing any of the endogenous SHR spatiotemporal expression dynamics, and does endogenous SHR also need to be expressed early during the cell cycle just before the asymmetric cell divisions? Or is endogenous SHR already expressed and has reached the presumed threshold levels by the start of this cell cycle? How does the exogenous overexpressed SHR levels compare to the endogenous SHR levels?"

Response: We agree that this is a critical point and was not sufficiently addressed in the submitted manuscript. We believe that our inducible SHR-GFP is functionally equivalent to endogenous SHR and that the model of SHR function that we have described here (low transient levels of SHR (and SCR) present early in the cell cycle are sufficient for asymmetric division) reflects the behavior of endogenous SHR for several reasons:

- In our inducible system, SHR actually *is* produced in its natural spatial range. SHR is induced transcriptionally in the SHR transcriptional domain (the central tissues of the root) and moves outward into the adjacent cell layer where it is sequestered in the nucleus and induces *SCR* transcription, just as in WT plants. We have moved the root diagram to Figure 1a and updated it to illustrate the behavior of endogenous and inducible SHR expression to clarify this point. Asymmetric divisions are produced farther up the root than they are in WT plants, but we believe that these cells are functionally equivalent to the wild-type CEID. They can divide asymmetrically, and we show in this new revision that they express *CYCD6* prior to those divisions (Extended Data Figure 2e and Extended Data Figure 8). *CYCD6* expression is normally restricted to the CEID in young roots (Sozzani *et al.*, 2010). We also show that *CYCD6* is *not* expressed prior to symmetric divisions in the *shr2* mutant (see Extended Data Figure 8).
- In this revision, we also now show that, in our inducible line, the levels of SHR-GFP present in the *shr* mutant layer at equilibrium, just prior to asymmetric division in fully induced (10 μ M dex) roots are similar to levels of SHR-GFP found in the CEI and CEID cells of WT plants (SHR:SHR-GFP *shr2*; Extended Data Figure 2d). When fully induced, SHR-GFP is indeed overexpressed in the central tissues of the root compared to WT, but the amount of SHR that enters the mutant tissue layer appears to be restricted to levels comparable to WT.
- We also tested conditions in which levels of SHR-GFP, even in the central tissues of the root, were much lower. We used different concentrations of dex (10, 1, 0.05, 0.03, 0.02, and 0.01 μ M) to induce SHR-GFP to different levels in order to determine the levels of SHR that are required for asymmetric division. The lower levels of dex (0.02 μ M and 0.03 μ M) produced very low levels of SHR. There was no overexpression anywhere in these roots, yet we still found asymmetric divisions.
- Regarding the temporal dynamics of SHR, it was not our intention to replicate the dynamics of endogenous SHR with the inducible system. The power of the system that we used is in our ability to generate a variety of dynamic expression profiles. Because only some of these resulted in asymmetric division, we were able to ask which features of these dynamics are critical to produce an asymmetric division. We could not have done this using WT plants, where the CEID always divides asymmetrically. The dynamics that we engineered are biologically relevant because we can rescue asymmetric division. We realize now that these points were not sufficiently clear in the submitted manuscript. We have revised the manuscript (see the first few paragraphs of the Results section) to highlight that we were not intending to capture the endogenous dynamics of SHR with the inducible line.
- We did measure the dynamics of endogenous SHR and SCR. In this revision, we have now quantified six time-courses that track the dynamics of SHR and SCR in the CEID from the time that the CEID is produced via anticlinal division of the CEI up to the time that the CEID divides asymmetrically (see Figure 3n and o and Supplemental Data File 3). We find that endogenous SHR is always present during the CEID cell cycle. It dips initially during division of the CEI, but never reaches zero, and

recovers quickly to CEI levels. Thus, endogenous SHR and SCR are likely always present during the CEID cell cycle, providing the conditions necessary for asymmetric division.

- Although the previous bistable model did not consider inducible SHR expression, we do make the reasonable assumption that if high stable steady state levels of SHR and SCR are required for asymmetric division in the WT context, they would also be required to rescue asymmetric divisions in the mutant. ODE models predict the behavior of a network (gene expression) as it moves toward equilibrium from a set of initial starting conditions. So, to test an ODE model, the steady state equilibrium of the network must be perturbed. Starting from 0 using an inducible system and observing how gene expression moves to its equilibrium state is a natural way to test an ODE model's predictions of the network's behavior.
- Our conclusions derived from experimentation with our inducible line are supported by previous research using endogenous SHR and SCR. These studies have shown that transient endogenous SCR expression (Heidstra, Welch and Scheres, 2004) as well as significantly reduced endogenous SCR expression (Cui *et al.*, 2007) are sufficient for asymmetric division. In addition, Koizumi *et al.* (2012) showed that high levels of SHR inhibit asymmetric division, and plant wounding studies have shown that reactivation of CYCD6 expression and periclinal division following plant wounding is preceded by a strong decrease in SHR expression (Marhava *et al.*, 2019). Furthermore, previous work in both plants and animals has suggested that division plane orientation is set in G1 (Tilman and Kimble, 2005; Facette, Rasmussen and Van Norman, 2019).

While we cannot be 100% sure that endogenous SHR and SCR operate similarly, the evidence we present is strong. Utilizing this inducible system enabled the discoveries that we made here. It would not have been possible to assess the relevance of various features of SHR and SCR dynamics, or their levels, by simply observing the dynamics of endogenous SHR and SCR. We also wish to point out that most of the research into the dynamics of cell cycle control has been performed in cell lines, which are essentially an artificial system to make experimentation feasible and takes the cells entirely out of the organism. A key advance of the work we present is that we can observe the behavior of two key regulators of cell division within single cells in an *in vivo* context.

Comment 6. *“Since the authors use an inducible overexpression system, and both SHR and SCR are transcription factors that need to modulate expression of target genes (some of them producing proteins that regulate cell cycle), it is not clear from the text if it is not kind of obvious that SHR and SCR would need to be expressed early in the cell cycle for their activities (which involve new protein synthesis of the downstream targets) to have any significant influence on cell-cycle progression and outcome of cell division.”*

Response:

The preprophase band (PPB), which is a band of microtubules encircling the nucleus in plant cells and marks the future plane of division, is not established until G2 (Facette, Rasmussen and Van Norman, 2019). If we consider the timescales involved for transcription and translation and the length of the cell cycle, it is not at all apparent that SHR and SCR would need to act early in the cell cycle. The average cell cycle length in our experiments is 10 hours. Taken from the Bio-numbers website (<https://bionumbers.hms.harvard.edu/search.aspx>), the average rate of transcription in eukaryotes is 2kb/min, while the average rate of translation is ~9AA/sec. The average gene length in Arabidopsis is 2.2kb and the median protein length is 356 AA. Thus, it takes approximately 1 minute to transcribe the average Arabidopsis gene and less than 1 min to translate it. On a timescale of 10 hours, SHR and SCR could act in G2 or late S phase and there would still be enough time to transcribe and translate its targets prior to PPB formation. We have now included these points in the Discussion.

Comment 7. *“The predictive statistical analysis of the data is nice, but it does not prove the requirement of certain SHR and/or SCR levels at a specific stage of the cell cycle. Could the overall relationship between SHR/SCR levels and asymmetric vs. symmetric division (or asymmetric division vs. non-division) decisions merely reflect the limitations in the speed of inducible expression of the overexpressed SHR protein? Again, it is not explained whether this artificial scheme recapitulate any of the endogenous SHR spatio-temporal expression patterns.”*

Response:

We apologize, but we are not sure what is meant by the “speed of inducible expression of the overexpressed SHR protein.” Does this refer to the diffusion rate of the inducible SHR-GFP into the mutant layer cells? In our inducible system, SHR-GFP is transcribed in the central tissues of the root, and SHR-GFP protein moves into the adjacent cell layer. In plants where SHR-GFP is driven by the endogenous SHR promoter (SHR:SHR-GFP), the same thing happens: SHR-GFP moves into the adjacent cell layer. Thus, movement of SHR-GFP protein from the central tissues of the root is independent of the method used to express it. We measured the rate of increase of SHR-GFP in the nuclei of the mutant cell layer and found a range of rates. These appeared to vary with the concentration of SHR present in the central tissues of the root, with the highest rates corresponding to positions with the highest SHR concentration in the center of the root. We showed earlier that SHR-GFP is overexpressed in the central tissues of fully induced (10 μ M dex) roots, but we tested dex concentrations all the way down to 0.01, where SHR-GFP is undetectable. Thus, our compendium of SHR trajectories in *shr* mutant cells should span the full range of possible diffusion rates in the endogenous context.

In this work, we do go beyond the predictive statistical analysis and perform experiments using hydroxyurea (in the previously submitted manuscript) and oryzalin (this revision) to confirm our in-silico-derived hypothesis. These experiments confirm that cell cycle stage is important in determining sensitivity to SHR. Cells are more likely to divide if they perceive SHR at an earlier stage of the cell cycle. These experiments utilize our inducible SHR, but we have many reasons to believe that endogenous SHR behaves in similar ways (see Response to Comment 5). Our finding that low levels of SHR and SCR are sufficient to trigger asymmetric division is consistent with previous findings that asymmetric divisions still occurred in a SCR RNAi transgenic line where SCR expression was reduced by 60% (Cui *et al.*, 2007) and the finding that division plane orientation is determined at the G1/S transition has been suggested by previous reports (Tilman and Kimble, 2005; Facette, Rasmussen and Van Norman, 2019). It has also been shown previously that transient expression of SCR can trigger asymmetric division (Heidstra, Welch and Scheres, 2004).

As mentioned above in Comment 5, we were not attempting to recapitulate the kinetics of endogenous SHR with our inducible system. Rather, we used this system to engineer a variety of kinetic profiles, and used those profiles to test which features of those kinetics are important for asymmetric division. However, we also did capture the endogenous SHR kinetics by measuring SHR-GFP and SCR-mKATE2 driven by their native promoters in plants with a wild-type phenotype (Figures 3n and o and Supplementary Data File 3). These kinetics vary widely, but in all 6 cases that we were able to capture, one constant is the presence of SHR and SCR early in the cell cycle of the CEID.

Comment 8. *“Finally, although the bistable switch model might not fit the data from the inducible SHR overexpression in the *shr2* background, the authors do not perform any additional experiments that can further elucidate new mechanisms – most of the discussion in the paper is speculative. I am not sure then if the general readership in Nature is thus left kind of unsatisfied.”*

Response: We do propose an alternative mechanism by which SHR and SCR trigger asymmetric division. Instead of the bistable switch, which requires high steady state levels of SHR and SCR, the new mechanism we propose is that the presence of SHR and SCR above a low threshold early in the cell cycle determines whether the cell will divide asymmetrically or symmetrically. The hydroxyurea experiments provide direct experimental evidence that position in the cell cycle is important for this decision, and that the critical window of sensitivity to SHR and SCR happens early in the cell cycle. We have strengthened these experimental findings in this revision by including experimental data where we used a G2/M cell cycle inhibitor, oryzalin, to show that exposing cells to SHR during a late phase of the cell cycle results in fewer asymmetric divisions. We have also included new images and corresponding histograms in Figure 3i – m to better emphasize this important experiment.

In the discussion and in Figure 4, the new model that we propose is based on the data that we presented in this work, as well as previously published data showing that:

- D-type cyclins act during G1 to phosphorylate RBR (Boniotto and Gutierrez, 2001)
- CYCD6;1/CDKB1 is an active complex that phosphorylates RBR (Cruz-Ramírez *et al.*, 2012) .
- RBR-associated cyclin-D/CDK activity peaks during the G1/S transition and early S phase (Boniotto and Gutierrez, 2001).
- SHR and SCR directly activate CYCD6 (Sozzani *et al.*, 2010).
- CYCD6 is expressed specifically in the CEID (Sozzani *et al.*, 2010).
- SHR and SCR act during G1 or early S to switch the orientation of the division plane (this work).
- Induction of SHR outside of the meristem is not sufficient to trigger cell division (this work).
- Low, transient levels of SHR and SCR in our inducible system can trigger asymmetric division (this work)
- Low levels of endogenous SCR can trigger asymmetric division (Cui *et al.*, 2007)
- Transient expression of endogenous SCR can trigger asymmetric division (Heidstra, Welch and Scheres, 2004)
- Induction of SHR in a *shr2* mutant activates *CYCD6* expression prior to asymmetric division (this revised work, see Response to Comment 5).
- Induction of SHR in a *shr2* mutant does not activate *CYCD6* prior to symmetric division (this revised work, see Response to Comment 5).

We have removed the RBR-E2F bistable switch (Yao *et al.*, 2008) from Figure 4b in order to better focus our new model on the findings of this work.

The advances of this work go beyond the biological insight into how two developmental regulators interface with the cell cycle. Many papers have been published over the last two decades extolling the potential for advances in imaging to elucidate biological mechanisms through quantification of protein dynamics in living organisms. These data are invaluable for testing existing mathematical models and can reveal truths that are very different from modeled mechanisms. Yet the end-to-end challenges associated with executing this type of research project are daunting, and require expertise in microscopy, developmental biology, computer science, and physics.

We were able to meet these challenges through an interdisciplinary effort that took over 10 years to complete. The light sheet microscope that we built allowed us to track multiple regulators in tandem, over long time-periods, in growing roots, in near-physiological conditions, and with high spatial and temporal resolution. The nature of the data we acquired fits into an imaging niche that, to our knowledge, hasn't been previously filled.

We feel the general readership of *Nature* will be left feeling satisfied with the combination of technical achievement, quality of the imaging, novel experimental design and approach, unique dataset of in-tandem transcription factor dynamics in single cells in a living organism, novel analysis approaches, along with experimentally-confirmed biological insight into a fundamental mechanism of cell cycle control.

Comment 9. *“Kinetics of gene expression measured using fluorescent protein reporters might need to be corrected for the maturation time of the fluorescent proteins. Experiments in E Coli show temperature dependent maturation rates of up to several hours, with an almost 2-fold increase when temperature drops from 37degC to 32 degC (Balleza, E., Kim, J. & Cluzel, P. Systematic characterization of maturation time of fluorescent proteins in living cells. Nat Methods 15, 47–51 <https://doi.org/10.1038/nmeth.4509> (2018). <https://doi.org/10.1038/nmeth.4509>). I wonder what GFP, mKate and RFP maturation times are at the experimental temperature of 21 degC used for the root imaging experiments in the manuscript. This might be important, since some of the results rely on accurate quantification of FP expression dynamics (e.g. Fig 2f-h).”*

We agree that it is important to consider the ramifications of delays due to maturation time kinetics on our conclusions. Taken from the paper that was suggested (Balleza, Kim and Cluzel, 2018), the typical maturation time (to 90% maximal fluorescence) is approximately 1 hour for eGFP and 2 hours for mKate2 at 32 degrees Celsius. Using the data presented in Balleza et al. (2018), we estimated the maturation time at 21 degrees Celsius to be approximately 4 hours to reach 90% (or approximately 2 hours as the typical time to 50% maturation) for mKate2, and approximately 2 hours to reach 90% (or approximately 1 hour to 50% maturation) for eGFP. These estimations were done by extrapolating the time to reach 90% of maximal fluorescence using the two temperatures that were previously published (Balleza, Kim and Cluzel, 2018) using an Arrhenius-like equation: $t_{90\%} = Ae^{-kT}$, where A is the pre-factor, T is the temperature, and k is the typical rate of change of the function with temperature. Because of the timescales involved (we are measuring changes in gene expression over the course of 24 to 48 hours), even at 21 degrees Celsius we predicted that our primary conclusions would not be impacted by correcting for maturation time.

To confirm this, we corrected our expression measurements for maturation time using a simple 1st order ODE model of the dynamics (Alber *et al.*, 2018):

$$\begin{aligned}\frac{dD(t)}{dt} &= P(t, \theta) - \alpha D(t) - \frac{1}{\tau} D(t) \\ \frac{dF(t)}{dt} &= \frac{1}{\tau} D(t) - \alpha F(t)\end{aligned}$$

where D(t) is the “dark” protein, P is the production term of the protein, where it degrades linearly with a rate of α and “lights up” with a rate of $1/\tau$. F is the amount of fluorescent protein.

Using the second equation and the estimated tau, we can extract the value for the amount of dark protein present. The total amount of protein is therefore:

$$T(t) = D(t) + F(t) = \tau \left(\frac{dF(t)}{dt} + \alpha F(t) \right) + F(t)$$

Therefore, up to the degradation rate, we can estimate the amount of total protein from the amount (intensity) that we measured. Having said that, this calculation uses the rate of change in time of the measurable protein – an operation that increases the amount of noise in the data. To reduce the noise, we used a spline interpolation of the data. An example for such a correction can be found in Extended Data Figure 9a. The main effect of the correction is to shift the TFs peak earlier (by $\sim 1/\tau$). The maturation-corrected data are noisier than the original data, but our main conclusions remain the same (Extended Data Figure 9). Because of the extensive data processing that this correction requires – we address this briefly in the methods and Extended Data Figure 9 – we have maintained the results in the main figures as they were submitted and present an example of the maturation-corrected data and the main conclusions/figures of the paper using the maturation-corrected data in Extended Data Figure 9.

Comment 10: *“Pg 10. “The lower prediction accuracy obtained using SCR levels suggests that its role in asymmetric division is secondary to that of SHR.” – It is not clear if the slight decrease in predictive power for SCR vs. SHR is biologically significant, or there are technical reasons for this different (e.g. use of mKate vs. GFP, different SNR in imaging the two channels, etc.)”*

Response: We agree and have removed this comment from the manuscript.

Comment 11: *“Pg 12-13 “To understand how our findings inform division of the CEID in wild-type plants ...” – this paragraph presents a single observation of one cell division in one experiment! It is not clear if it is appropriate to draw any conclusions from just n=1 observation, and if a more controlled experiment and appropriate statistical analysis should be performed. Otherwise this paragraph and related discussions should be removed.”*

In this revision, we have added 5 additional time courses that contain a complete CEID cell cycle, from the time of CEI division when the CEID is formed, up to the time that the CEID divides asymmetrically. It was particularly challenging to capture these two rare divisions together in one 48-hour time course. We imaged over 70 roots in order to capture these 6 time courses.

Comment 12: *“Some of the experimental parameters are not clear. E.g., Fig 1e-f, Suppl Videos 5&6 – what is the concentration of dex used? The text only specifies “low dex” induction. Similarly, I could not find in the text the dex induction levels for the data used to construct the predictive models. Are these analyses mostly relying on the “fully induced” 10 uM dex conditions or on the lower (0.02-0.05 uM dex) induction conditions?”*

Response: We have added to the text and legends the specific parameters, including the dex concentration, used for each experiment where this information was missing. For Figure 1e (now 1f) the dex concentration used was 0.02 μM . For Figure f (now e) we used 0.02 and 0.03 μM . For the ODE fittings (Figures 2e-h) we used all 40 μM dex experiments. We used the entire dataset (all dex concentrations) for the machine learning algorithms.

Comment 13: *“The manuscript could be improved by better annotation of some of the figures and, especially, the videos.”*

Response: Thank you for this suggestion. We have added annotations to the figures and videos where possible. We think the manuscript is much improved and easier to follow with these additions.

Comment 13: “Perhaps the authors could better explain how the image acquisition and analysis pipeline in Ext Data Fig 3c relates to the corresponding Methods sub-sections.”

Response: Thank you for pointing this out. We agree that the workflow diagram in Extended Data Figure 3c and the Methods text didn’t match up very well. We have modified both the workflow diagram in the figure (now Extended Data Figure 4) and the Methods text to better align the two.

Comment 14: “Ext Data Fig 4 – x axes needs labels.”

Response: We have added labels to the x axes in this figure (now Extended Data Figure 7).

Comments from Referee #3:

“The article entitled “Patterning and growth are coordinated early in the cell cycle” aims to provide a mechanistic understanding of how two transcription factors contribute to the decision-making between formative and proliferative division in a cell cycle-dependent manner. The authors used a custom-made light-sheet imaging system to perform quantitative 4D and 3D imaging in vivo combined with mathematical models based on simple ODE.”

Comment 1: “The title is too generic, mainly because the manuscript focuses on only two transcription factors; a specific title would be more appropriate for this manuscript.”

Response: We have changed the title: to “SHORTROOT and SCARECROW coordinate patterning and growth early in the cell cycle.”

We do believe, however, that the findings we share here likely apply to other asymmetric divisions and developmental regulators in other systems. The RB/CYCLIND/E2F pathway in humans is deeply conserved and is more similar to plants than to yeast or fungi (Zluhan-Martínez *et al.*, 2020; Shimotohno *et al.*, 2021). D-type cyclins have been implicated in axis determination in metazoans such as *C. elegans* (Tilman and Kimble, 2005), in addition to their known roles in promoting cell cycle progression. It is established that coordination of axis determination and cell-cycle progression by G1/S regulators are important for asymmetric division in metazoans (Tilman and Kimble, 2005; Noatynska *et al.*, 2013; Costa, 2017). Thus, our findings may point to a shared mechanism used to coordinate axis and cell fate determination (patterning) with cell cycle progression (growth) across eukaryotic systems.

Comment 2: “The authors first report the function of SHR in controlling asymmetric cell division through imaging roots expressing an inducible SHR line in vivo using a light sheet confocal microscope that allowed extended live imaging without bleaching or phototoxicity. While the imaging quality is superb, the outcome is not novel as the function of SHR using this inducible SHR line is well established in the host laboratory, yet, they were useful for the modeling section.”

Response: The first demonstration that SHR induction in a *shr2* mutant background can rescue asymmetric divisions was by Sozzani *et al.* (2010) using a SHR:SHR-GR *shr2* transgenic line. This provided induction of activity through interaction with the GR moiety. Our inducible SHR line is novel in that SHR is induced at the transcriptional level and labeled with GFP (SHR:GAL4-GR UAS:SHR-GFP *shr2*), allowing us to use imaging to directly observe movement of this mobile transcription factor from one tissue type to another, and to quantify those dynamics for analysis. All the conclusions that we made in this work

were based on these measured dynamics. We verified that Sozzani et. al. (2010) is properly cited when introducing the inducible SHR construct.

Comment 3. *“The authors based their analysis on the inducible SHR (SHR:GAL4-GR UAS:SHR-GFP). However, while looking at the confocal images, although the line complemented the shr mutants, the expression level in the endodermis was very low when compared to images in previous studies from the Benfey’s laboratory (<https://www.hhmi.org/content/benfey-research-abstract-slideshow>), Nakajima et al., 2001; Cui et al., 2007, etc.). Since the whole manuscript is based on imaging and quantifying SHR levels, the differences in some conclusions with the previously published bistable model might be caused by the differences in SHR levels in the endodermis between the lines used in the two different studies. It is well plausible that the conclusions from this manuscript are valid but performing similar experiments with another inducible line where the SHR promoter could be fused for instance, to an estrogen receptor-based transactivator XVE to drive the SHR protein and then use this line to monitor the SHR dynamics and assess whether the same results are obtained. That would strengthen challenging the previously published model”.*

Response: We appreciate this concern and realize that we didn’t address this apparent issue in the original manuscript. In plants expressing SHR-GFP from the SHR promoter, levels of SHR-GFP in the CEI and CEID are lower than in the endodermis (Nakajima et al., 2001; Cui et al., 2007). The mutant tissue layer cells in our study are more comparable to these CEI and CEID cells than to endodermal cells. In this revision, we were able to directly quantify levels of SHR-GFP in mutant layer cells of fully induced (10 μ M dex) roots just prior to asymmetric division. Levels of SHR-GFP in the CEI and CEID cells in the non-inducible line (SHR:SHR-GFP shr2) are comparable to and fall within the broader range of SHR levels that were tested using the rescued line (Extended Data Figure 2d). After division of the mutant tissue layer cells in our inducible system, SHR-GFP levels increase in the resulting endodermis, just as they do in SHR:SHR-GFP plants. Thus, we feel that the results we obtained are unlikely to be due to differences in levels of SHR.

We did make estradiol-induced SHR-GFP transgenic plants in response to this comment, but unfortunately obtained very low levels of SHR expression from the lines that we generated and were unable to perform experiments with it. We have updated the text in the manuscript and added a new extended data figure (Extended Data Figure 2d) to clarify that the low levels of SHR in the mutant tissue layer prior to rescued asymmetric division are comparable to levels of SHR-GFP in the CEID prior to asymmetric division in SHR:SHR-GFP shr2 plants. It is also worth noting that our purpose in using the inducible line was to be able to modulate levels of SHR to assess the importance of SHR expression level in the decision to divide asymmetrically. One of our key findings is that high levels of SHR are not needed for asymmetric division to occur.

Comment 4. *“Furthermore, the work published by Heidstra et al. in 2004 (Mosaic analyses using marked activation and deletion clones dissect Arabidopsis SCARECROW action in asymmetric cell division) provides excellent tools to study both SHR and SCR roles in controlling the divisions and assess whether these are symmetric or asymmetric. Combining the SCR activation and deletion clones with the inducible SHR in one plant and taking advantage of the light sheath technology and the modeling will provide more substantial evidence and robust data to support the observations obtained in this manuscript.”*

Response: This is an interesting suggestion, but unfortunately it would not be possible to complete this experiment within a reasonable timeframe. (Moreover, the activation and deletion clones were created nearly 20 years ago, so it is unlikely they could be obtained.) If we understood the suggestion correctly,

this would involve incorporating the clonal activation/deletion system in a *shr* mutant with inducible SHR. Because the clones are formed somewhat randomly, this would involve imaging a very large number of plants to capture instances of activation or deactivation during the time in which SHR is induced.

It is notable, however, that in this paper that the authors also found evidence that transient SCR activity was sufficient for asymmetric division.

Comment 5. *“The authors also provide a new model for SHR and SCR to control asymmetric cell division by regulating CYCD6 while the symmetric division is through other cyclins. They state that SHR and SCR would activate CYCD6 during the G1 phase of the cell cycle. However, these observations were based on treatment with the cell cycle inhibitor HU, and there was no genetic data to support these conclusions. One way to validate the outcome of this study would be to do these experiments using the CYCD6 marker and show that it is activated only during asymmetric cell division.”*

Thank you for this suggestion. We agree that it is important to show that CYCD6 is activated specifically during asymmetric division, and not during symmetric division, in support of the model that we present in Figure 4. In this revision we have added these data to Extended Figure 2e (showing that SHR activates CYCD6 prior to asymmetric division), and Extended Data Figure 8 (showing that CYCD6 is not activated during symmetric divisions). We also point out that it was previously shown that CYCD6 is specifically activated during asymmetric, and not symmetric divisions in wild-type plants (Sozzani *et al.*, 2010).

The model in Figure 4 is based on the following data from both this work (now updated) and previous studies:

- SHR and SCR act during G1 or early S to switch the orientation of the division plane (this work).
- Induction of SHR outside of the meristem is not sufficient to trigger cell division (this work).
- SHR and SCR directly activate CYCD6 (Sozzani *et al.*, 2010).
- CYCD6 is expressed specifically in the CEID (Sozzani *et al.*, 2010).
- Induction of SHR activates CYCD6 prior to asymmetric division (this revised work).
- Induction of SHR does not activate CYCD6 prior to symmetric division (this revised work).
- D-type cyclins act during G1 to phosphorylate RBR (Boniotto and Gutierrez, 2001)
- CYCD6;1/CDKB1 is an active complex that phosphorylates RBR (Cruz-Ramírez *et al.*, 2012) .
- RBR-associated cyclin-D/CDK activity peaks during the G1/S transition and early S phase (Boniotto and Gutierrez, 2001).

Comment 6. *“The authors should also use other cell cycle inhibitors (G2-M phase) to prove that the action is G1-S specific.”*

Response: Thank you for this good suggestion. In this revision, we used a cell cycle inhibitor specific to G2/M to synchronize the cell cycle prior to induction of SHR expression. While treatment with the G1/S inhibitor, hydroxyurea, resulted in a larger number of asymmetric divisions (94% asymmetric compared to 50% in untreated roots), treatment with the G2/M inhibitor, oryzalin, resulted in many fewer asymmetric divisions (20%) after SHR induction. These new data further support our hypothesis that SHR is required at an early stage of the cell cycle by showing that cells at a later stage are not competent to respond to SHR. These data are now incorporated into the text and Figure 3i-m.

Comments from Referee #4:

“In this work authors address the regulation and coordination between formative (asymmetric) and proliferative (symmetric) cell divisions, a key question in developmental biology. They analyze in detail these two types of cell divisions in the stem cell niche of the Arabidopsis root using light sheet and confocal microscopy in an inducible system of SHR, which together with SCR, are involved in asymmetric cell division (ACD). Their main claim is that the existing bistable model is challenged. Indeed they obtained results supporting the conclusion that this model does not apply and instead, low and transient levels of SHR are required for ACD. They conclude this based on detailed measurements of SHR kinetics. They also propose that expression of SHR early in the cell cycle determines the type of cell division.

The work is technically challenging and authors master the experimental strategies used. The main claims of this study, namely, the nature of SHR as a driver and that low and transient levels of SHR are sufficient to determine ACD, are supported by the data provided, although some loose ends remain.”

Comment 1. *“A major point not sufficiently addressed (and discussed) is whether the known role of CYCD6;1, and other components of the pathway in CEID division, applies to other cells developing ACD.”*

We agree that we should have addressed this point in the Discussion. We have now included the following text:

SHR, SCR, and CYCD6 are expressed in cell types other than the CEID that also undergo asymmetric division, such as developing lateral root primordia, vascular tissues including the pericycle and phloem, and bundle sheath cells in leaves (De Smet *et al.*, 2008; Sozzani *et al.*, 2010; Lucas *et al.*, 2011; Cui *et al.*, 2014; Kim *et al.*, 2020). Each of these tissues displays patterning defects in *shr* mutants associated with defective asymmetric divisions. This suggests the SHR/SCR/CYCD6 pathway may regulate other formative divisions within the plant. More broadly, D-type cyclins have also been implicated in axis determination in metazoans such as *C. elegans* (Tilman and Kimble, 2005), in addition to their known roles in promoting cell cycle progression. It is established that coordination of axis determination and cell-cycle progression by G1/S regulators are important for asymmetric division in metazoans (Tilman and Kimble, 2005; Noatynska *et al.*, 2013; Costa, 2017). Thus, our findings may point to a shared mechanism used to coordinate axis and cell fate determination (patterning) with cell cycle progression (growth) across eukaryotic systems.

We also added this point:

The RB/CYCLIND/E2F pathway in humans is deeply conserved and is more similar to plants than to yeast or fungi (Zluhan-Martínez *et al.*, 2020; Shimotohno *et al.*, 2021).

Comment 2. *“Also, why does an ACD is followed by a symmetric division and not by new ACDs, without restrictions (since the inducible strategy is producing SHR in a continuous manner)?”*

Although SHR appears to be overexpressed in the central root tissues in the inducible construct, we show in this revision that the amount of SHR protein present in the mutant tissue layer prior to asymmetric division is similar to the amount of SHR present in the CEID cell in WT roots (see Extended Data Figure 2D). Thus, it is likely that the influx of SHR into the mutant tissue layer cells in the *shr* mutant is similar to SHR influx into the CEID cell in WT roots. There is evidence that suggests that the higher levels of SHR present after ACD inhibit further cell divisions (Koizumi *et al.*, 2012). Notably, levels of SHR

are higher after asymmetric division in the resulting endodermal cells in our inducible system, just as they are in WT roots (see Figure 2c, 40 hr timepoint, green channel image; also Figure 1c, 20 hr timepoint, green channel image).

Comment 3. *“A general comment is that it is not easy to follow and find all the necessary information in the figure legends. In Fig. 1b, the focus is on 5 distal cells. Later in the text, the term “all cells” is frequently mentioned. Do authors refer to “all these five cells” or to “all cells in the meristem file”, or else?”*

Response: The cell file in Figure 1b (now 1d) was intended to be a representative cell file. We have made sure that this is reflected in the legend. We also revised all of the figure legends and text for clarity and incorporated additional information where the experimental parameters may not have been clear. We have also added a few annotations to the figures to make it easier to find key information.

Comment 4. *“Fig. 1d. This figure is not sufficiently described in the text. This panel should include the dividing cell at the end (12h?).”*

Response: We have moved this figure to Extended Data Figure 1a and provided a better description, along with the image of the dividing cell at the end.

Comment 5. *“There is an apparent decrease of SHR amount at ~6h. Is this observed in other cells analyzed? Does this have a relevance for ACD? In fact, the accumulation pattern seems to be different in different cells: maximum levels reached towards the end of the cell cycle (cell #5), a peak early in the cell cycle and then steady levels (cell #2), a peak and then a decrease (cells #1 and #4). Please comment.”*

Response: The power of our dex induction system is that it allowed us to generate different dynamic patterns of SHR expression. Since not all these expression patterns resulted in asymmetric division, this allowed us to identify the features of SHR expression that are important for asymmetric division. Since many cells divide without the dip at 6 hours, we conclude that this feature of this particular cell is not relevant for determining whether the cell will divide asymmetrically. We have clarified these points in the text in the first paragraphs of the Results section.

Comment 6. *“SHR sharply disappear coinciding with mitosis. Is this the results of targeted proteolysis and/or dilution out of chromatin?”*

Cruz-Ramirez et al. (2012) also observed a similar decrease in SHR and SCR expression at mitosis. Through treatment with MG132, an inhibitor of the 26S proteasome, they observed an increase in SCR-GFP expression, but not SHR-GFP, suggesting that SCR is subject to proteasome-mediated degradation, but that the reduction of SHR expression appears to be the result of other factors. Dilution out of chromatin also likely plays a role in the decrease in expression observed at mitosis, since the total volume occupied by the available protein rapidly increases when the nuclear envelope breaks down.

Comment 7. *“Fig. 3f. The prediction accuracy is close to >90% for smaller cells. However, it is also quite high (~50%) for cells with larger sizes. Does this mean that the proposed relevance of SHR level early in the cell cycle is important but that high levels later in the cell cycle also lead to ACD?”*

We agree that this is a confusing point. The approximately 50% prediction accuracy is what we would expect to obtain by chance. This analysis is equivalent to predicting the accuracy of a coin toss. Each

time you make a prediction for how the coin will land, and you will be right 50% of the time because there is equal probability that the coin will land heads or tails. To improve this analysis, we have added a statistical test to show which results are statistically different from what would be expected by chance using a binomial test.

In this particular analysis, if SHR or SCR crosses the threshold in the specified nuclear size window, the cell is predicted to divide. The prediction accuracy is the percentage of cells that cross the threshold in that size window that do divide, or that don't cross the threshold and don't divide (so where we get this prediction right). The prediction accuracies in the 0.25 window are significantly greater than what is expected by chance ($P = 3.8 \times 10^{-91}$, 8.9×10^{-61} , 8.0×10^{-43} for Confocal - SHR, LS - SHR, and LS - SCR, respectively), which means that if SHR crosses the threshold in that 0.25 window and we predict it will divide, we will be right more than 50% of the time, because our hypothesis that SHR triggers asymmetric cell division during that window is predictive.

The bar chart (Figure 3e) now contains the p-values from the binomial tests that we performed, and we added a note in the figure legend that states that an approximate 50% accuracy is expected by chance. We hope that these results make more sense now. Thank you very much for your close attention to this figure and for identifying this subtle point. It likely would have been confusing for other readers as well, and we think including the binomial test results will clarify the figure.

Comment 8. *“Ext. Data Fig 2i. Cells completing a cell cycle in such a short time, both asymmetric and symmetric cell divisions, is unprecedented in the literature. This is also difficult to reconcile with results shown in Fig 3k, in the absence of hydroxyurea, when a high fraction of cells respond to SHR dividing asymmetrically (and therefore, responding early in the cell cycle, as proposed here). Such high fraction is suggestive of a longer (early) cell cycle, otherwise the window of opportunity to respond to SHR would be much smaller.”*

Response:

We found a small bug in the code and after fixing it, the median cell cycle times are slightly higher, with a median cell cycle time around 8 and 11 hours, for asymmetrically and symmetrically dividing cells, respectively. Rahni and Birnbaum (2019) found a median cell cycle time in the meristem for light grown roots to be 12.57 hours. We used the data from Rahni and Birnbaum (2019) (Additional File 16) and calculated the median cell cycle time specifically for endodermal cells in positions 2-5 (Position 0 is the QC. Positions 2-5 are comparable to the first five cells up from the QC that we included in our analysis.) and found it to be 12 hours.

Our finding of a median cell cycle duration of 8 and 11 hours, then, is not very different from the 12 hours measured (Rahni and Birnbaum, 2019). However, this was only a minor point and isn't central to our conclusions, so we have removed it from the Discussion and the extended data figure.

You mentioned that the window of opportunity would need to be long for such a high percentage of cells to be able to respond to SHR and divide asymmetrically in the absence of hydroxyurea. We would like to point out that the 25th percentile of the nuclear size range often corresponds to a larger percentile of the time range of the cell cycle. Please see Figure 3e for a visualization of this. The nucleus often increases in size only gradually in the beginning of the cell cycle and then increases more rapidly later (presumably due to DNA synthesis during S phase). For the cell shown in Figure 3e, by the end of the 25th percentile of the nuclear size range, the cell cycle is already almost half-way completed. It

seems reasonable then that ~50% of the cells would be in a responsive cell cycle phase when exposed to SHR and would divide asymmetrically.

Comment 9. *“Induction of CYCD6;1 expression is a requirement to determine ACD in the stem cell. Authors do not show whether this is also the case in the cells upon SHR expression is induced. Does ACD in cells (other than the stem cell) depends on CYCD6;1? Is the auxin balance relevant for the ACDs in these cells?”*

We agree that it is important to show that CYCD6 is induced prior to rescue of asymmetric divisions in the *shr2* mutant. This is needed to confirm that the rescued asymmetric divisions are comparable to asymmetric division of the CEID, which is controlled by the SHR/SCR/CYCD6 network. We have now added these data to Extended Data Figure 2e (using a comparable SHR induction system, SHR:SHR-GR). Another reviewer pointed out that we should also show that CYCD6 is *not* induced prior to symmetric division. We confirmed this and added these data to Extended Data Figure 8.

Cruz-Ramirez et al. (2012) showed that treatment with exogenous auxin can trigger extra asymmetric divisions along the length of the root, and can potentiate CYCD6 expression after SHR induction in a *shr2* mutant background (SHR:SHR-GR *shr2*), an experimental system that is comparable to our inducible SHR system (SHR:GAL4-GR UAS:SHR-GFP *shr2*). Induction of SHR:SHR-GR also rescues asymmetric divisions absent in the *shr* mutant. Thus, auxin appears to play a role in these asymmetric divisions. Understanding the role of auxin in ACD, however, was not the focus of our study, and isn't relevant to the conclusions we reach about SHR and SCR.

There are likely many other players involved in making the decision to divide asymmetrically. SHR and SCR have many other direct targets, including the BIRD transcription factors (Bolle, 2016) and other cell cycle regulators (Sozzani *et al.*, 2010). In this study, we chose to focus on SHR and SCR specifically, as well as their only confirmed direct link to the cell cycle, CYCD6.

Comment 10. *“Fig. 4. Showing direct measurements of SHR accumulation in newly-formed cells after a symmetric cell division would be a strong support to the main claim. This will also provide a more precise identification of when SHR level is critical early in the cell cycle.”*

We agree that showing a quantification of SHR in newly formed cells after symmetric division of the CEI would support our claim that SHR is present early in the cell cycle of the CEID. We now include this as part of Figure 3o and include data from an additional 5 CEI/CEID cells in Supplemental Data File 3. From this quantification it is apparent that SHR expression never decreases to zero during division of the CEI. Thus, it is present early in the CEID cell cycle and is able to direct the cell to divide asymmetrically.

We also show quantification of SHR after a symmetric division in our inducible line in Figure 1d, cell 4 as well as in the Supplemental Data Files 1 and 2 for the confocal and light sheet data, respectively. In Figure 1d, the symmetric division is indicated with the black dotted line, while asymmetric divisions are indicated with the orange dotted lines.

Comment 11. *“What is the expression pattern of cell fate genes after Dex treatment?”*

Sozzani et al. (2010) performed a time course microarray experiment after dex treatment of a SHR:SHR-GR *shr2* inducible line. The endodermal cell fate regulators MGP, NUC, SCL3, and SCR were induced between 3 – 12 hours after dex treatment. The changes in gene expression observed in this experiment

after dex treatment are likely very similar to the changes that occur in response to our inducible SHR-GFP (SHR:GAL4-GR UAS:SHR-GFP). **Redacted**

References:

- Alber, A.B. *et al.* (2018) 'Single Live Cell Monitoring of Protein Turnover Reveals Intercellular Variability and Cell-Cycle Dependence of Degradation Rates', *Molecular Cell*, 71(6), pp. 1079-1091.e9. Available at: <https://doi.org/10.1016/j.molcel.2018.07.023>.
- Balleza, E., Kim, J.M. and Cluzel, P. (2018) 'Systematic characterization of maturation time of fluorescent proteins in living cells', *Nature Methods*, 15(1), pp. 47–51. Available at: <https://doi.org/10.1038/nmeth.4509>.
- Bolle, C. (2016) 'Chapter 19 - Functional Aspects of GRAS Family Proteins', in D.H. Gonzalez (ed.) *Plant Transcription Factors*. Boston: Academic Press, pp. 295–311. Available at: <https://doi.org/10.1016/B978-0-12-800854-6.00019-1>.
- Boniotti, M.B. and Gutierrez, C. (2001) 'A cell-cycle-regulated kinase activity phosphorylates plant retinoblastoma protein and contains, in Arabidopsis, a CDKA/cyclin D complex', *The Plant Journal*, 28(3), pp. 341–350. Available at: <https://doi.org/10.1046/j.1365-313X.2001.01160.x>.
- Costa, S. (2017) 'Are division plane determination and cell-cycle progression coordinated?', *The New Phytologist*, 213(1), pp. 16–21.
- Cruz-Ramírez, A. *et al.* (2012) 'A bistable circuit involving SCARECROW-RETINOBLASTOMA integrates cues to inform asymmetric stem cell division', *Cell*, 150(5), pp. 1002–1015. Available at: <https://doi.org/10.1016/j.cell.2012.07.017>.
- Cui, H. *et al.* (2007) 'An evolutionarily conserved mechanism delimiting SHR movement defines a single layer of endodermis in plants', *Science*, 316(5823), pp. 421–425. Available at: <https://doi.org/10.1126/science.1139531>.
- Cui, H. *et al.* (2014) 'SCARECROW, SCR-LIKE 23 and SHORT-ROOT control bundle sheath cell fate and function in Arabidopsis thaliana', *The Plant Journal: For Cell and Molecular Biology*, 78(2), pp. 319–327. Available at: <https://doi.org/10.1111/tpj.12470>.
- De Smet, I. *et al.* (2008) 'Receptor-Like Kinase ACR4 Restricts Formative Cell Divisions in the Arabidopsis Root', *Science*, 322(5901), pp. 594–597. Available at: <https://doi.org/10.1126/science.1160158>.
- Desvoyes, B. *et al.* (2020) 'A comprehensive fluorescent sensor for spatiotemporal cell cycle analysis in Arabidopsis', *Nature Plants*, 6(11), pp. 1330–1334. Available at: <https://doi.org/10.1038/s41477-020-00770-4>.
- Facette, M.R., Rasmussen, C.G. and Van Norman, J.M. (2019) 'A plane choice: coordinating timing and orientation of cell division during plant development', *Current Opinion in Plant Biology*, 47, pp. 47–55. Available at: <https://doi.org/10.1016/j.pbi.2018.09.001>.

Heidstra, R., Welch, D. and Scheres, B. (2004) 'Mosaic analyses using marked activation and deletion clones dissect Arabidopsis SCARECROW action in asymmetric cell division', *Genes and Development*, 18(16), pp. 1964–1969. Available at: <https://doi.org/10.1101/gad.305504>.

Kim, H. *et al.* (2020) 'SHORTROOT-Mediated Intercellular Signals Coordinate Phloem Development in Arabidopsis Roots[OPEN]', *The Plant Cell*, 32(5), pp. 1519–1535. Available at: <https://doi.org/10.1105/tpc.19.00455>.

Koizumi, K. *et al.* (2012) 'The SHORT-ROOT protein acts as a mobile, dose-dependent signal in patterning the ground tissue', *Proceedings of the National Academy of Sciences*, 109(32), pp. 13010–13015. Available at: <https://doi.org/10.1073/pnas.1205579109>.

Lucas, M. *et al.* (2011) 'SHORT-ROOT Regulates Primary, Lateral, and Adventitious Root Development in Arabidopsis1[C][W][OA]', *Plant Physiology*, 155(1), pp. 384–398. Available at: <https://doi.org/10.1104/pp.110.165126>.

Marhava, P. *et al.* (2019) 'Re-activation of Stem Cell Pathways for Pattern Restoration in Plant Wound Healing', *Cell*, 177(4), pp. 957–969. Available at: <https://doi.org/10.1016/J.CELL.2019.04.015>.

Nakajima, K. *et al.* (2001) 'Intercellular movement of the putative transcription factor SHR in root patterning', *Nature*, 413(6853), pp. 307–311. Available at: <https://doi.org/10.1038/35095061>.

Noatynska, A. *et al.* (2013) 'Coordinating cell polarity and cell cycle progression: what can we learn from flies and worms?', *Open Biology*, 3(8), p. 130083. Available at: <https://doi.org/10.1098/rsob.130083>.

Rahni, R. and Birnbaum, K.D. (2019) 'Week-long imaging of cell divisions in the Arabidopsis root meristem', *Plant Methods 2019 15:1*, 15(1), pp. 1–14. Available at: <https://doi.org/10.1186/S13007-019-0417-9>.

Shimotohno, A. *et al.* (2021) 'Regulation of the Plant Cell Cycle in Response to Hormones and the Environment', *Annual Review of Plant Biology*, 72, pp. 273–296. Available at: <https://doi.org/10.1146/annurev-arplant-080720-103739>.

Sozzani, R. *et al.* (2010) 'Spatiotemporal regulation of cell-cycle genes by SHORTROOT links patterning and growth', *Nature [Preprint]*. Available at: <https://doi.org/10.1038/nature09143>.

Tilman, C. and Kimble, J. (2005) 'Cyclin D Regulation of a Sexually Dimorphic Asymmetric Cell Division', *Developmental Cell*, 9(4), pp. 489–499. Available at: <https://doi.org/10.1016/j.devcel.2005.09.004>.

Yao, G. *et al.* (2008) 'A bistable Rb-E2F switch underlies the restriction point', *Nature Cell Biology*, 10(4), pp. 476–482. Available at: <https://doi.org/10.1038/ncb1711>.

Zluhan-Martínez, E. *et al.* (2020) 'Beyond What Your Retina Can See: Similarities of Retinoblastoma Function between Plants and Animals, from Developmental Processes to Epigenetic Regulation', *International Journal of Molecular Sciences*, 21(14), p. 4925. Available at: <https://doi.org/10.3390/ijms21144925>.

Reviewer Reports on the First Revision:

Referees' comments:

Referee #2 (Remarks to the Author):

The revised manuscript is better written and organized compared to the original submission. The comparison with predictions of the bistable model is improved, and helps convey the new findings and the limitations of the inducible system as well as the imaging plus mathematical modeling approach.

I have some suggestions for further improvements:

Lines 112-114 - "Studies using MS2 and similar systems allow for the imaging of transcription factor dynamics at the level of single molecules^{38,39}". MS2 system images nascent transcription at single-gene level, and usually does not quite achieve (RNA) single-molecule resolution. Single transcription factor, as well as single-molecule RNAP Polymerase II imaging is achieved by more sensitive optical imaging systems, e.g. PMID: PMC6675578 DOI: 10.1016/j.cell.2019.05.029.

Line 131 - "long timescales" - define "long"; e.g. hr-long, day-long, etc.

Line 206-208 - "These SHR kinetics are inconsistent with any biologically relevant bistable model in which high steady state levels of nuclear SHR are necessary to trigger division." - Define "biologically relevant", and give references or point to other analyses done in the paper to reach this conclusion.

Extended Data figures do not have titles.

Ext Data Fig 1b - is the dark green line the trace corresponding to the figures? What are the light green lines?

The authors now have captured $n=6$ instead of $n=1$ trajectories of SHR-GFP and SCR-mKATE2 dynamics with a CEI division and a sub-sequent CEID division, but only show one such trace. They should show all 6 traces. Do all 6 traces show the features described (fluctuations in SHR levels, quick recovery to pre-division levels)? Is there more information in those traces that can be extracted with further analysis (e.g. by time-aligning and averaging them?)

Ext Data Fig 5a-c - The authors choose to show dR 10, pS2 3, and dS 10, but only dS 10 fits the data reasonably well. There are other examples that fit the data better than dR10 and pS2 3 (e.g. pS1 0.033 and pS1 0.01).

Referee #3 (Remarks to the Author):

The authors have addressed all my comments and I only have a one minor suggestion.

The model in Figure 1b and Figure 4b, RBR does not repress SCR but binds to it and therefore prevents its association to SHR, please correct this in the figure, the inhibition sign should point to the SCR-SHR complex.

Referee #4 (Remarks to the Author):

We appreciate the significant efforts made by the authors to address our comments, which we believe were constructive. We are satisfied with their response and by including new data that clarify the text and further support their claims.

Referee #5 (Remarks to the Author):

The manuscript authored by Winter et al., offers a quantitative analysis of the actions of two transcription factors involved in controlling formative cell divisions. The authors are commended for their efforts in employing cutting-edge microscopy techniques to provide essential time-lapse imaging analyses, which shed light on the dynamics of plant development. However, from my expertise in modeling, I have observed that while the methods utilized are appropriate, the models themselves are rather basic and do not consider the spatial aspects of root growth and tissue mechanics. Incorporating these elements could significantly enhance the impact of author's work. Notably, there are existing model frameworks that enable the simulation of root growth and/or biochemical processes (i.e., Mahonen & Ten Tusscher et al, 2014; Marconi et al., 2021 among others), which could be explored to improve the study.

I have a several suggestions and comments regarding the current version of the manuscript:

- 1) The use of the term "asymmetric cell division" might be misleading, as cells technically divide symmetrically, albeit with a flipped division plane. To avoid confusion, I recommend using the term "formative cell divisions" instead, particularly to aid readers from non-plant backgrounds.
- 2) Upon reviewing the time-lapse data presented in Fig. 1d, I observed an intriguing transient damped oscillatory dynamic. This raises questions about the dependence of SHR on cell cycle regulators. It would be valuable to explore whether SHR acts downstream of a bistable/oscillatory system within the cell cycle itself. Addressing and discussing this point in the manuscript would strengthen its findings.
- 3) In Figure 2d, the authors utilized a relatively high concentration of DEX (40 μ M) without providing controls where plants were treated with DEX at this concentration alongside the WT scenario SCR:SCR-mKATE2 marker. It is well-known that such high DEX concentrations can significantly impact plant growth and patterning. Additionally, the profile of 'd' (I assume for $n = 274$ cells) appears markedly different from 'e-h.' However, it remains unclear whether these data points originate from the same roots or different experiments.
- 4) Extended Data Figure 2b reveals intriguing pulse-like dynamics at low DEX levels (0.05 μ M), which appear to be less toxic. These dynamics may support the original bistable model, but the current version of the manuscript does not include any discussion on this matter.
- 5) Finally, the biggest achievement of this work could be to explain how exactly these formative divisions are oriented, while it is clear SCR controls CYCD6, it is puzzling what flips the division plane. Manuscript would benefit from discussions on this intriguing matter. Is there any connection

between transient SHR dynamics observed for lower Dex concentrations and division plane Flipping mechanism perhaps through (microtubules, CLASP proteins?).

Author Rebuttals to First Revision:

Response to Referees

Comments from Referee #2:

Comment 1:

“The revised manuscript is better written and organized compared to the original submission. The comparison with predictions of the bistable model is improved, and helps convey the new findings and the limitations of the inducible system as well as the imaging plus mathematical modeling approach.

I have some suggestions for further improvements:

Lines 112-114 - “Studies using MS2 and similar systems allow for the imaging of transcription factor dynamics at the level of single molecules^{38,39}.” MS2 system images nascent transcription at single-gene level, and usually does not quite achieve (RNA) single-molecule resolution. Single transcription factor, as well as single-molecule RNAP Polymerase II imaging is achieved by more sensitive optical imaging systems, e.g. PMID: PMC6675578 DOI: 10.1016/j.cell.2019.05.029.”

Response:

We removed this sentence from the text.

Comment 2. *“Line 131 – “long timescales” – define “long”; e.g. hr-long, day-long, etc.”*

Response: We removed this sentence from the text.

Comment 3. *“Line 206-208 – “These SHR kinetics are inconsistent with any biologically relevant bistable model in which high steady state levels of nuclear SHR are necessary to trigger division.” – Define “biologically relevant”, and give references or point to other analyses done in the paper to reach this conclusion.”*

Response: We have changed the text to read: “These SHR kinetics are inconsistent with a bistable model in which high steady state levels of nuclear SHR are necessary to trigger division, where the scale of the model is comparable to the observed range of SHR protein levels (Figure 1e-g).”

Comment 4. *“Extended Data figures do not have titles.”*

Response: We have added titles to the Extended Data figures.

Comment 5. *“Ext Data Fig 1b – is the dark green line the trace corresponding to the figures? What are the light green lines?”*

Response: Yes, the dark green line corresponds to the cell highlighted in the images. We removed the lighter green lines (these were redundant with Current Revision Figure 1f) and modified the legend for clarity.

Comment 6. *“The authors now have captured $n=6$ instead of $n=1$ trajectories of SHR-GFP and SCR-mKATE2 dynamics with a CEI division and a sub-sequent CEID division, but only show one such trace. They should show all 6 traces. Do all 6 traces show the features described (fluctuations in SHR levels, quick recovery to pre-division levels)? Is there more information in those traces that can be extracted with further analysis (e.g. by time-aligning and averaging them?)”*

Response: We have added 5 new trajectories to Extended Data Figure 8. All 6 traces show fluctuating levels of SHR and SCR that remain detectable throughout the time courses.

Comment 7. *“Ext Data Fig 5a-c – The authors choose to show $dR 10$, $pS2 3$, and $dS 10$, but only $dS 10$ fits the data reasonably well. There are other examples that fit the data better than $dR10$ and $pS2 3$ (e.g. $pS1 0.033$ and $pS1 0.01$).”*

Response: We have added the examples $pS1 = 1/10$, $1/30$, and $1/100$ to Extended Data Figure 5a-c.

Comments from Referee #3:

Comment 1: *“The authors have addressed all my comments and I only have a one minor suggestion. The model in Figure 1b and Figure 4b, RBR does not repress SCR but binds to it and therefore prevents its association to SHR, please correct this in the figure, the inhibition sign should point to the SCR-SHR complex.”*

Response: We have corrected this in Figure 1b and Figure 4b.

Comments from Referee #4:

“We appreciate the significant efforts made by the authors to address our comments, which we believe were constructive. We are satisfied with their response and by including new data that clarify the text and further support their claims.”

Response: No further action is required.

Comments from Referee #5:

Comment 1: *“The manuscript authored by Winter et al., offers a quantitative analysis of the actions of two transcription factors involved in controlling formative cell divisions. The authors are commended for their efforts in employing cutting-edge microscopy techniques to provide essential time-lapse imaging analyses, which shed light on the dynamics of plant development. However, from my expertise in modeling, I have observed that while the methods utilized are appropriate, the models themselves are rather basic and do not consider the spatial aspects of root growth and tissue mechanics. Incorporating these elements could significantly enhance the impact of author’s work. Notably, there are existing model frameworks that enable the simulation of root growth and/or biochemical processes (i.e., Mahonen & Ten Tusscher et al, 2014; Marconi et al., 2021 among others), which could be explored to improve the study.”*

Response: Thank you for your comment. We do agree that the models we present are basic, but this point supports our argument that the more complex bistable model is not needed to explain the regulatory relationship between SHR and SCR. We considered the possibility of adding a spatial dimension to the modeling. In order to obtain a proper comparison, we would need to apply both the bistable and our new model to a spatial model of a growing root that incorporates the *shr* mutant phenotype and rescue of divisions. We estimate that in order to do that we would need to

have a model with a few hundred ODEs. This exercise increases the complexity of the modeling and it is not clear how it would help to increase our understanding. Implementation of a spatial model is not a trivial task and would require careful consideration of the parameters to include, and many rounds of optimization. Unfortunately, it would not be possible to implement such a model in the timeframe we have (4 weeks) for the revision. It is possible to do this if we have more time, but again, it is not evident how this addition would improve upon our findings.

Comment 2: *“The use of the term “asymmetric cell division” might be misleading, as cells technically divide symmetrically, albeit with a flipped division plane. To avoid confusion, I recommend using the term “formative cell divisions” instead, particularly to aid readers from non-plant backgrounds.”*

Response: We have changed the term “asymmetric division” to “formative division” and “symmetric division” to “proliferative division” throughout the manuscript.

Comment 3: *“Upon reviewing the time-lapse data presented in Fig. 1d, I observed an intriguing transient damped oscillatory dynamic. This raises questions about the dependence of SHR on cell cycle regulators. It would be valuable to explore whether SHR acts downstream of a bistable/oscillatory system within the cell cycle itself. Addressing and discussing this point in the manuscript would strengthen its findings.”*

Response:

The oscillations mentioned in this comment are indeed very intriguing. We characterized them throughout the data, and the typical periodicity for these oscillations is 3 ± 1 for the confocal data and 4.5 ± 2 for the light sheet. We can observe about 2 to 3 oscillations per cell cycle.

Redacted

Comment 4: *“In Figure 2d, the authors utilized a relatively high concentration of DEX (40 μ M) without providing controls where plants were treated with DEX at this concentration alongside the WT scenario SCR:SCR-mKATE2 marker. It is well-known that such high DEX concentrations can significantly impact plant growth and patterning.”*

Response: We now include a control showing that SHR:SHR-GFP SCR:SCR-mKATE2 UBQ10:H2B-CFP *shr2* roots grown on 40uM dex are correctly patterned (Extended Data Figure 10c).

Comment 5: *“Additionally, the profile of 'd' (I assume for $n = 274$ cells) appears markedly different from 'e-h.' However, it remains unclear whether these data points originate from the same roots or different experiments.”*

Response: We have updated the figure legend to clarify that the SHR and SCR trajectories shown in 'd' are from a single representative cell from a fully induced root (treated with 40uM dex), while the trajectories shown in 'e-h' are averages of all cells from all fully induced roots.

Comment 6: *“Extended Data Figure 2b reveals intriguing pulse-like dynamics at low DEX levels (0.05 μ M), which appear to be less toxic. These dynamics may support the original bistable model, but the current version of the manuscript does not include any discussion on this matter.”*

Response: We apologize, but it is unclear to us how the pulse-like dynamics at low dex levels could support the bistable model. In fact, we use these dynamics to support the claim that low transient levels of SHR can trigger formative divisions. Our hypothesis for the origin of these dynamics is that, due to the instability of dex, at such low concentrations the level of dex in the media falls below the threshold required for activation of SHR transcription after some period of time. Thus, *SHR-GFP* expression is lost in the central tissues of the root, SHR-GFP protein can no longer move into the ground tissue, and the SHR-GFP already present in the ground tissue degrades through normal degradation processes.

Comment 7: *“Finally, the biggest achievement of this work could be to explain how exactly these formative divisions are oriented, while it is clear SCR controls CYCD6, it is puzzling what flips the division plane. Manuscript would benefit from discussions on this intriguing matter. Is there any connection between transient SHR dynamics observed for lower Dex concentrations and division plane Flipping mechanism perhaps through (microtubules, CLASP proteins?).”*

Response: We agree that this is an exciting avenue for future study. **Redacted**

Reviewer Reports on the Second Revision:

Referees' comments:

Referee #2 (Remarks to the Author):

The authors have meaningfully addressed most of the previous comments. The revised sentence "These SHR kinetics are inconsistent with a bistable model in which high steady state levels of nuclear SHR are necessary to trigger division, where the scale of the model is comparable to the observed range of SHR protein levels (Figure 1e-g)." and Figure 1e-g are still not very clear. What is meant by "the scale of the model"? This is a rather vague term. Does it refer to temporal scale/dynamics, absolute expression levels, something else?

Perhaps Fig. 1e should have additional details (e.g. showing a cell division, labeling/annotating the x/time axis etc.). Are we supposed to compare Figure 1e to Figure 1f and Figure 1d, and what is supposed to be different? One would expect the idealized step function to be rounded in a real, experimental system. It is not clear at all that real, experimental curves like the ones in Figures 1d,f are inconsistent with the idealized curve in Fig 1e; perhaps the authors should better highlight what real features, if any, are different than Fig 1e.

Referee #5 (Remarks to the Author):

I agree with authors that introducing spatial aspects in the model would require more time, and I hope authors consider doing so in the future as this would add next level of understanding to their proposed mechanism.

Regarding, pulsatile dynamics of SHR/SCR if it is indeed driven by cell cycle, SHR/SCR may exhibit characteristics of bistability or oscillations which is the underlying dynamics of cell cycle machinery. As authors suggested this is potential venue for investigation that will help in the future to prove or falsify proposed models.

In summary I am happy to recommend this manuscript for publication, as this work opens an intriguing venues for future research.

Author Rebuttals to Second Revision:

Response to Referees

Comments from Referee #2:

Comment 1:

The authors have meaningfully addressed most of the previous comments. The revised sentence “These SHR kinetics are inconsistent with a bistable model in which high steady state levels of nuclear SHR are necessary to trigger division, where the scale of the model is comparable to the observed range of SHR protein levels (Figure 1e-g).” and Figure 1e-g are still not very clear. What is meant by “the scale of the model”? This is a rather vague term. Does it refer to temporal scale/dynamics, absolute expression levels, something else?

Response:

We have added the text in red below in order to clarify this point:

“These SHR kinetics are inconsistent with a bistable model in which high steady state levels of nuclear SHR are necessary to trigger division, where the scale of the model (**predicted SHR levels**) is comparable to the observed range of SHR protein levels (Figure 1e-g).”

Comment 2:

Perhaps Fig. 1e should have additional details (e.g. showing a cell division, labeling/annotating the x/time axis etc.). Are we supposed to compare Figure 1e to Figure 1f and Figure 1d, and what is supposed to be different? One would expect the idealized step function to be rounded in a real, experimental system. It is not clear at all that real, experimental curves like the ones in Figures 1d,f are inconsistent with the idealized curve in Fig 1e; perhaps the authors should better highlight what real features, if any, are different than Fig 1e.

Response:

We have added the words ‘low’ and ‘high’ to Figure 1e to highlight the 2 steady states in the predicted SHR curve, as well as the words ‘low peak’ to the plots in Figure 1f to highlight the feature of these measured SHR curves that differs from the predicted low and high steady states.

Reviewer Reports on the Third Revision

Referee #2

(Remarks to the Author)

I appreciate the authors' responses to the previous comments, and have no further comments.